# Leishmania survives by exporting miR-146a from infected to resident cells to subjugate inflammation

Satarupa Ganguly[1,*], Bartika Ghoshal[1,*], Ishani Banerji[1,2,*], Shreya Bhattacharjee[1,2], Sreemoyee Chakraborty[1,2], Avijit Goswami[1], Kamalika Mukherjee[1], Suvendra N Bhattacharyya[1,2]

***Leishmania donovani,*** the causative agent of visceral leishmaniasis, infects and resides within tissue macrophage cells. It is not clear how the parasite infected cells crosstalk with the non-infected cells to regulate the infection process. During infection, *Leishmania* adopts a dual strategy for its survival by regulating the intercellular transport of host miRNAs to restrict inflammation. The parasite, by preventing mitochondrial function of host cells, restricts the entry of liver cell derived miR-122–containing extracellular vesicles in infected macrophages to curtail the inflammatory response associated with miR-122 entry. On contrary, the parasite up-regulates the export of miR-146a from the infected macrophages. The miR-146a, associated with the extracellular vesicles released by infected cells, restricts miR-122 production in hepatocytes while polarizing neighbouring naïve macrophages to the M2 state by affecting the cytokine expression. On entering the recipient macrophages, miR-146a dominates the miRNA antagonist RNA-binding protein HuR to inhibit the expression of proinflammatory cytokine mRNAs having HuR-interacting AU-rich elements whereas up-regulates anti-inflammatory IL-10 by exporting the miR-21 to polarize the recipient cells to M2 stage.

## Introduction

*Leishmania donovani* (*Ld*) is the causative agent of visceral leishmaniasis that affects a large portion of the human population in the Indian subcontinent and also in sub-Saharan Africa (Lenk et al, 2018). The apicomplexan parasite has a dual-stage life cycle and lives in the gut of the sandfly vector as promastigotes. The promastigote changes to amastigote stage after entering the mammalian host macrophage cells (Sunter & Gull, 2017). The promastigotes enter the mammalian host with the saliva of the sandfly vector introduced during the blood meal and subsequently make entry into the hepatic tissue where they infect the Kupffer cells at the initial stage of infection before the infection load is transferred to the spleen (Walker et al, 2014). The *Ld* parasite lives in the tissue macrophages within a specialized subcellular structure called parasitophorous vacuole and alters the signalling components of the infected macrophages to polarize to M2 stage. *Ld* ensures low expression of proinflammatory cytokines like IL-1β and TNF-α, whereas the anti-inflammatory cytokines IL-10 and IL-4 expression get enhanced in the infected host (Mukherjee et al, 2013). The status of the noninfected macrophages present in the infection niche is not clear, but it is certain that the noninfected macrophages present in the infection niche should not get activated to ensure the overall dominance of an anti-inflammatory response that needs to be maintained in the infected tissue. How the parasite, which remains within the specialized vacuole structure of the infected host macrophage, cross-communicates with the resident noninfected macrophages to suppress expression of inflammatory cytokines is an important question to explore. In this context, the extracellular signals derived from resident noninfected macrophages and hepatocytes that can activate the infected macrophage should also be counteracted within the infected host cell milieu.

Extracellular vesicles (EVs) are released by different types of mammalian cells that are used primarily by mammalian immune cells to cross-communicate the cellular status across cell boundaries (Regev-Rudzki et al, 2013; Fernandez-Messina et al, 2015). It is known that the infected cell EVs can be used to transfer the parasite derived and host factors to target specific pathways in neighbouring cells to ensure establishment of systematic infection and its propagation (Regev-Rudzki et al, 2013; Kalluri & LeBleu, 2020). miRNAs are important post-transcriptional regulators of gene expression that are expressed in a cell type and stage specific manner in mammalian hosts (Bartel, 2018). miRNAs are also known to be communicated across the cell boundary to affect neighbouring cell fates. Therefore, if communicated from the infected host cells, miRNAs, as an epigenetic signal, could alter the gene expression process in neighbouring cells. The role of specific miRNAs in regulation of expression of pro or anti-inflammatory pathway components

[1]RNA Biology Research Laboratory, Molecular Genetics Division, Council of Scientific and Industrial Research (CSIR)-Indian Institute of Chemical Biology, Kolkata, India
[2]Academy of Scientific and Innovative Research (AcSIR), CSIR-Human Resource Development Centre, (CSIR-HRDC) Campus, Ghaziabad, India

Correspondence: mb.kamalika@gmail.com; suvendra@iicb.res.in
*Satarupa Ganguly, Bartika Ghoshal, and Ishani Banerji contributed equally to this work.

in mammalian immune cells has already been studied (Lindsay, 2008).

In this article, we have documented an extraordinary mechanism that the internalized *Ld* adopts to ensure an anti-inflammatory infection milieu in the infected liver of the mammalian host. The internalized parasite prevents the proinflammatory response in the infected macrophage by preventing the entry of hepatocyte derived miR-122–containing EVs. The parasites achieve it through the depolarization of mitochondria of the host cells by enhancing the expression of the uncoupler protein Ucp2 that restricts the entry of the hepatic EVs into the infected cells and prevents miR-122 induced inflammatory responses. We have also noted miR-146a upregulation in the infected cells and the excess miR-146a also gets packaged into the EVs released by the infected macrophages. EVs containing miR-146a enter the hepatocytes to restrict the production of miR-122 there. The miR-146a–containing EVs are also internalized by the noninfected macrophages that then get polarized to the M2 stage to express IL-10 in a miR-146a dependent manner. HuR, the RNA binding protein with known miRNA antagonistic function, induces export of miR-21 out of macrophage cells to cause high expression of miR-21 target IL-10 in the noninfected cells in presence of miR-146a. Interestingly, HuR itself gets eventually repressed by high miR-146a. This in turn, reduces the HuR-mediated stabilization and enhanced expression of ARE-containing proinflammatory cytokines in the recipient neighbouring noninfected macrophage cells. Thus, by cross communicating the infected host cell–derived miR-146a, *Leishmania* ensures its own survival by regulating both miR-122 and HuR in neighbouring hepatocyte and naïve macrophage cells, respectively.

# Results

### Secretion of miR-122 by activated hepatocytes

Lipopolysaccharide or LPS is an immunogen, derived from the cell wall of the Gram-negative bacteria and is known to stimulate the macrophage cells via activation of TLR4 receptor and p38/MAPK pathway (Bode et al, 2012). LPS increases the expression of proinflammatory cytokines by enhancing the NF-κβ–dependent transcription and also by inactivating the repressive miRNAs in LPS-activated macrophages (Mazumder et al, 2013). In mammalian liver, the LPS stimulation leads to activation of the tissue resident macrophages, and thus, LPS increases liver inflammation (Rex et al, 2019). Hepatocytes also respond to LPS and altered metabolic function of the hepatocytes exposed to LPS has been documented (Masaki et al, 2004; Momen-Heravi et al, 2015). miR-122 is the key pro-inflammatory miRNA expressed in hepatocytes (Momen-Heravi et al, 2015). We documented decreased miR-122 level in mammalian liver cell Huh7 exposed to LPS (Fig S1A and B). The decreased miR-122 level was associated with enhanced phospho-p38 MAPK level in LPS-treated Huh7 cells (Fig S1C). The decreased cellular miR-122 was associated with increased miR-122 detected in the EVs released by LPS-treated Huh7 cells (Fig S1D and E). What consequence could this EV-associated miR-122 have on the tissue resident macrophages infected with *Leishmania*? Upon infection

of the host, the liver is the first tissue where the *Ld* initiates the infection process by targeting the tissue macrophage the Kupffer cells (Beattie et al, 2010). Hepatic miR-122, secreted as part of EVs from hepatocytes, can interact with macrophages to transfer the miRNA to the resident macrophages and could enhance the expression of pro-inflammatory cytokines (Momen-Heravi et al, 2015). Therefore, the *Ld*-infected macrophages must adopt strategies to combat this activation process to protect themselves from getting killed by the immunostimulatory effect of hepatocyte secreted miR-122.

### *Leishmania* prevent internalization of proinflammatory miR-122 to the infected cells

*Ld* is known to affect miRNA machineries of host cells to ensure its proliferation (Ghosh et al, 2013; Chakrabarty & Bhattacharyya, 2017). Interestingly, like what happens in LPS-activated cells, expression of miR-122 also caused an increase in the TNF-α mRNA and protein levels in RAW264.7 macrophage cells (Fig 1A and C). Expression of iNOS and NO were also getting increased with miR-122 expression in macrophages (Fig 1C). Conversely, when *Ld* infection of RAW264.7 cells expressing miR-122 was followed, we found increased proinflammatory cytokine TNF-α and low anti-inflammatory cytokine IL-10 mRNA levels upon miR-122 expression there (Fig 1D and E). The infection level of respective cells was also found to be reduced when the macrophages received the miR-122 positive EVs before the infection (Fig 1F and G). Thus, the hepatic miR-122 has an immuno-protective role, and when transferred via EVs released by hepatic cells could prevent infection of neighbouring liver macrophage cells—the first target of invading *Ld* pathogen in the mammalian host (Ghosh et al, 2013; Momen-Heravi et al, 2015; Chakrabarty & Bhattacharyya, 2017). To survive, *Ld* must prevent this EV-mediated miRNA transfer process to stop inflammatory response in the host cells upon its interaction with miR-122 positive EVs. We hypothesized that *Ld* could have hijacked the inflammatory machinery of the host cell by preventing the miR-122–containing EV transfer to infected macrophages, and thus could ensure survival of the internalized pathogen. Consistent with the assumption, we found an increased production of proinflammatory cytokines and miR-155, the hallmark of inflammatory response in macrophage cells upon treatment with miR-122 positive EVs. But in cells already infected with *Ld*, the uptake of miR-122–containing EVs was found to be significantly compromised with concomitant reduction in miR-155 or TNF-α production in the infection context (Fig 1H). miR-122–containing EV treatment, however, do not have any major effect on cellular NO and iNOS level (Fig 1I).

### *Leishmania* inhibits entry of miR-122–containing EVs in the infected Kupffer cells in mouse liver

EVs packed with miR-122 were generated from mouse liver cells HePa1-6. The miR-122–containing EVs were used to treat the mouse primary macrophage to score the effect of *Ld* infection on internalization of miRNA in infected mouse primary macrophages (Fig 2A). We documented a substantial reduction in EV-mediated miRNA entry in infected cells (Fig 2B). To score the same in vivo, we used mice infected with *Ld* and measured the effect of infection

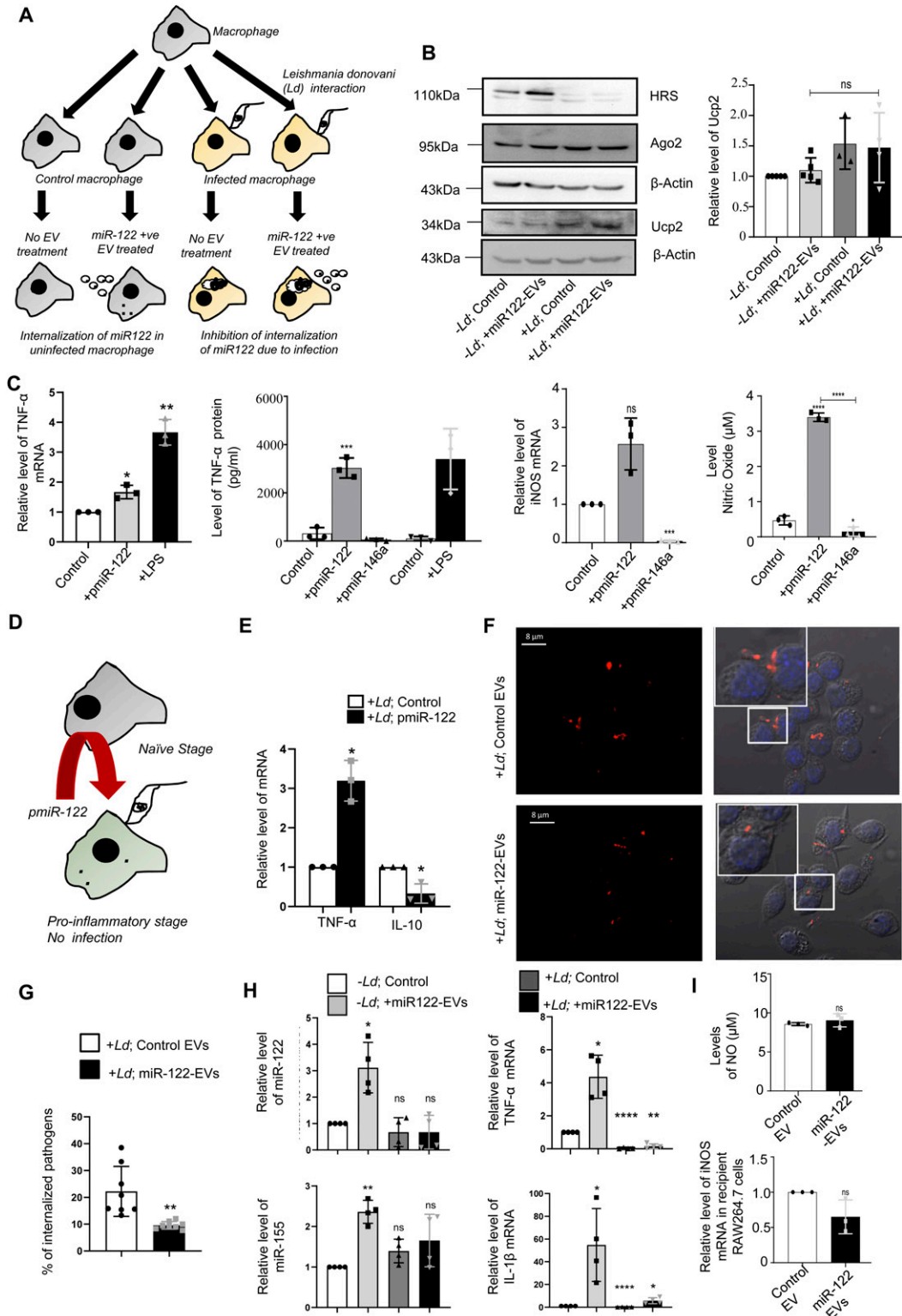

**Figure 1. Hepatic miR-122 acts as an immunostimulant for mammalian macrophage and *Ld* restricts miR-122 entry in infected macrophage.**
**(A)** Diagrammatic depiction of experiments done with *Leishmania donovani (Ld)*–infected or uninfected RAW264.7 cells treated with miR-122–containing extracellular vesicles (EVs). The macrophage cells were either infected or noninfected with *Ld* for 6 h after which both the groups were either treated with miR-122–containing EVs or left untreated. **(B)** Western blot analysis of RAW264.7 cells to show the infection related up-regulation of Ucp2 and down-regulation of HRS protein expression in RAW264.7 cells (*left panel*). Graphical representation of densitometric analysis of Ucp2 Western blot (*right panel*) (n ≥ 3 independent experiments, unpaired *t* test, *P* = 0.2139). Values for uninfected and untreated cells were set as unit. **(C)** Effect of miR-122 overexpression on cytokine expression and nitric oxide generation in RAW264.7 cells.

on EV-derived miR-122 entry in infected mouse liver macrophage cells (Fig 2C). After infection of 30 d and 24 h of injection of miR-122–containing EVs, the hepatocytes and Kupffer cells were separated on a Percoll gradient and enrichment of each cell population was monitored by following the expression of mRNAs encoding the marker genes (Fig 2D). The levels of infection in both control and EV-injected population were monitored by following the kinetoplast DNA content (Fig 2E). In the uninfected Kupffer cells, there was internalization of miR-122 after treatment with miR-122–containing EVs and a substantial increase in the expression of pro-inflammatory TNF-α mRNA in miR-122 EV-treated cells was noted (Fig 2F and G). However, the Ld infection substantially reduces the entry of miR-122 EVs in infected mouse Kupffer cells as both miR-122 and TNF-α expression was found to be substantially low in infected state (Fig 2H and I).

### *Leishmania* depolarizes mitochondria to prevent the entry of miRNA-containing EVs

Ld is known to cause a robust change in endocytic pathway of the host cells and alter components that are also found to be used for endocytic entry or exit of miRNAs (Lievin-Le Moal & Loiseau, 2016; Chakrabarty & Bhattacharyya, 2017). Proximity of endosomes with Ld-containing parasitophorous vacuoles in infected macrophage cells was noted (Fig S2A and B). The endosome maturation process in Ld-infected cells is known to get affected (Scianimanico et al, 1999). We explored the status of Rab proteins in Ld-infected macrophages and observed a decrease in endosomal protein Hepatocyte growth factor-regulated tyrosine kinase substrate (HRS) expression in infected cells (Fig 1B). However, there has not been a major change in Rab5a or RILP expression with Ld infection of RAW264.7 cells. Interestingly, treatment of cells with EVs containing miR-122 has no significant effect on endosome number or Rab5a expression in infected RAW264.7 cells. The infection also had no effect on Dynamin 2 expression and thus cannot account for the

reduced miR-122 entry that is known to be a dynamin 2 dependent process in mammalian cells (Fig S2B and C; [Ghoshal et al, 2021]).

It is known that Ld targets the mitochondrial dynamics and activity by inducing depolarization of mitochondria and reducing the mitochondria–ER–endosome interaction in infected cells (Chakrabarty & Bhattacharyya, 2017). Mitochondrial uncoupler protein Ucp2 gets up-regulated upon Ld infection (Fig 1B) suggesting a depolarized mitochondrial state in infected cells (Chakrabarty & Bhattacharyya, 2017). Does mitochondrial depolarization affect EV-entry? Mitochondria can interact with several cellular compartments, like ER and endosomes (Klecker et al, 2014; Todkar et al, 2019). The Ucp2 is a mitochondrial uncoupling protein causing defects in oxidative phosphorylation and ATP synthesis by changing membrane potential across mitochondrial membranes. Hence, it was anticipated that uncoupling of the mitochondrial potential can affect the internalization of EV-derived miRNAs in mammalian cells as it is known to affect endogenous miRNA activity (Chakrabarty & Bhattacharyya, 2017). FH-Ucp2 was expressed in the recipient HeLa cells and FH-Ucp2–expressing cells were incubated with miR-122–containing EVs (isolated from HeLa cells expressing pmiR-122). Ucp2 over-expression caused a disruption in the mitochondrial structures as observed microscopically (Fig 3A). The low level of internalization and consequently a lower repression activity of the transferred EV-derived miRNAs were observed in FH-Ucp2–expressing recipient HeLa cells (Fig 3B–D). It has been reported earlier that Ld infection or the loss of mitochondrial membrane potential by FH-Ucp2 are accompanied by reduced juxtaposition of ER and mitochondria (Chakrabarty & Bhattacharyya, 2017) (Fig S3A and B). Interestingly, oligomycin treatment that affects the ATP concentration alone and FCCP treatment that disrupts membrane potential of mitochondria without much effect on cellular ATP content, do not have any notable inhibitory effect on EV-mediated miR-122 entry in recipient cells (Fig S3C). These data suggest involvement of altered mitochondrial dynamics and mitochondrial interaction of subcellular organelles in FH-Ucp2–expressing cells to

---

The RAW264.7 macrophage cells were either kept as control, transfected with miR-122 expression plasmid pmiR-122 or with miR-146a expression plasmid pmiR-146a or treated with LPS (as positive control) to determine the mRNA level of TNF-α (C, *left panel*, n = 3 independent experiments; P = 0.0361, 0.0084) and iNOS (C, *middle right panel*, n = 3 independent experiments; P = 0.0562, 0.0002) from cellular RNA. Protein level of TNF-α (C, *middle left panel*, n = 3 independent experiments; unpaired *t* test, *P* = 0.0006) and nitric oxide level (*right panel*, n = 3 independent experiments; unpaired *t* test, P ≤ 0.0001, 0.0405, <0.0001) was also measured from culture supernatant. Quantification of TNF-α mRNA levels by qRT-PCR was done for the conditions described above. 18s RNA or GAPDH mRNA was used as endogenous control for qRT-PCR. Values for control plasmid transfected untreated cells were considered as unit for qRT-PCR (control). **(D)** Effect of miR-122 expression on Ld infection. This panel shows the schematic model for RAW264.7 cells transfected with miR-122 expression plasmid (pmiR-122) followed by L. donovani infection. **(E)** Comparison of the proinflammatory TNF-α, and anti-inflammatory IL-10 cytokine expression in pmiR-122 transfected or pCIneo control vector transfected RAW264.7 cells followed by infection with Ld. An increase in the proinflammatory cytokine TNF-α (P = 0.0180) and a decrease in the anti-inflammatory cytokine IL-10 (P = 0.0422) were observed in the presence of miR-122 (n = 3 independent experiments).18s rRNA was used as endogenous control. Values for pCIneo control transfected and infected RAW264.7 cells were considered as unit. **(F)** Microscopic analysis of Ld infection in RAW264.7 cells treated or untreated with miR-122–containing EVs. Cells were then visualized under the confocal microscope. The Leishmania protein GP63 imaged with indirect fluorescence (red) and cells with internalized parasites were counted. Scale bar 8 μm. Marked areas are zoomed for 5×. **(G)** Effect of miR-122–containing EVs on Ld infection. Graphical representation of the percent of RAW264.7 cells infected with L. donovani as observed microscopically (n = 132, number of cells; P = 0.0038, unpaired *t* test). **(H)** Quantification of miRNA levels upon Ld infection followed by EV treatment in RAW264.7 cells. Real-time PCR showed no increase in miR-122 levels in infected macrophages incubated with miR-122 positive EVs (*top left panel*; P = 0.0215, 0.3211, 0.3808). miR-155 levels were also quantified under similar conditions (*bottom left panel*; P = 0.0024, 0.0736, 0.1368) (n = 4 independent experiments). U6 was used as endogenous control. Relative levels of cytokine mRNA were also quantified. The proinflammatory cytokine, TNF-α (*top right panel*; P = 0.0142, <0.0001, 0.0013) and IL-1β (*bottom right panel*; P = 0.0438, <0.0001, 0.0366) did not increase in presence of the parasite followed by EV treatment (n = 4 independent experiments). 18s rRNA was used as endogenous control. Values for uninfected cells were set as unit. **(I)** Quantification of iNOS mRNA level and Nitric oxide (NO) generation after miR-122–containing EV treatment of RAW264.7 cells. Nitric oxide level was determined from culture supernatant of EV-treated RAW264.7 cells (*upper panel*, n = 3 independent experiments, unpaired *t* test; P = 0.3878). qRT-PCR analysis was used to determine the relative level of iNOS mRNA in recipient RAW264.7 cells (*lower panel*, n = 3 independent experiments; P = 0.1275). GAPDH mRNA was used as endogenous control. Values for control EV-treated cells were considered as unit for qRT-PCR. Data information: In all experimental data, error bars are represented as mean ± SEM, ns, nonsignificant, *P < 0.05, **P < 0.01, ***P < 0.001, ****P < 0.0001, respectively. P-values were calculated by two-tailed paired *t* test in most of the experiments unless mentioned otherwise. Positions of molecular weight markers are marked and shown with the respective Western blots. Source data are available for this figure.

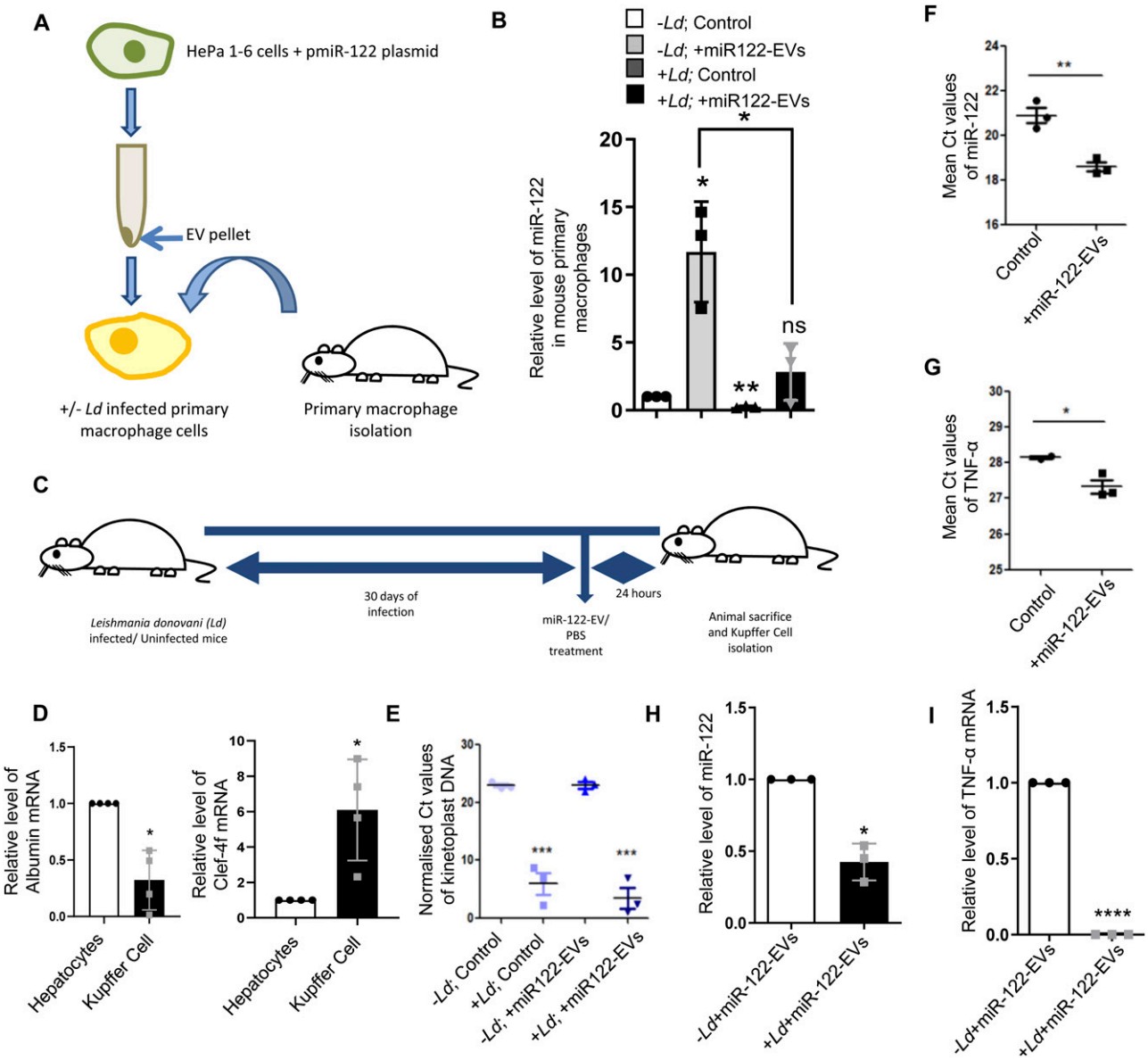

**Figure 2. miRNA uptake is prevented in *Ld*-infected primary macrophages and Kupffer cells.**
**(A)** A model to show the effect of *Leishmania* infection on uptake of miR-122–containing extracellular vesicles (EVs) in primary peritoneal macrophages. Macrophages were either infected or noninfected with *Ld* followed by miR-122–containing EV treatment. The EVs were obtained from miR-122 expressing murine hepatic cell HePa 1–6. **(B)** Relative levels of internalized miR-122 in infected or noninfected primary macrophages (n = 3 independent experiments; 0.0379, 0.0033, 0.2726, Unpaired *t* test *P* = 0.0227). U6 was used as endogenous control and values for uninfected control cells were considered as unit. **(C)** Schematic representation of *Ld* infection of mice followed by miR-122–containing EV treatment. Mice were infected with *Ld* for 30 d followed by tail vein injection of miR-122–containing EVs. After 24 h of treatment, mice were euthanized and Kupffer cells were isolated to quantify the miR-122 content. **(D)** Characterisation of isolated hepatocytes and Kupffer cells. Quantification of hepatocyte specific albumin mRNA in hepatocytes and Kupffer cells to show their low levels in Kupffer cell isolate (*left panel*; *P* = 0.143, n = 4 number of mice). Relative levels of macrophage specific C-type Lectin Domain Family 4, Member F (Clec4f) mRNA between hepatocytes and Kupffer cells showing high levels in Kupffer cells (*right panel*; *P* = 0.0378, n = 4 number of mice). GAPDH was used as endogenous control. Values for hepatocytes were considered as unit. **(E)** Comparison of normalized Ct values of Kinetoplast DNA in infected or uninfected Kupffer cells isolated from animals untreated or treated with EVs containing miR-122 to determine the parasitic load in Kupffer cells (n = 3 number of mice; *P* = 0.0009, 0.0005, unpaired *t* test). GAPDH was used as endogenous control. **(F)** Mean Ct values of internalized miR-122 in uninfected Kupffer cells either untreated or treated with miR-122–containing EVs (n = 3 number of mice; *P* = 0.0049, unpaired *t* test). **(G)** Mean Ct values of TNF-*α* mRNA in uninfected Kupffer cells upon treatment with miR-122–EVs compared with untreated Kupffer cells (n ≥ 2; *P* = 0.0444, unpaired *t* test). **(H)** The internalization of EV-derived miR-122 declined in the presence of *Leishmania* infection. Real-time PCR revealed the relative levels of miR-122 in uninfected or infected Kupffer cells isolated from animals treated with miR-122–containing EVs (n = 3 number of mice; *P* = 0.0162). U6 was used as endogenous control and values for uninfected miR-122 EV-treated cells were considered as unit. **(I)** Relative levels of proinflammatory cytokine TNF-*α* in *Leishmania* infected or uninfected miR-122–EV-treated animals' Kupffer cells (n = 3 number of mice; *P* < 0.0001). GAPDH mRNA was used as endogenous control and values for uninfected miR-122 EV-treated set were considered as unit. Data information: In all experimental data, error bars are represented as mean ± SEM, ns, nonsignificant, *P* < 0.05, **P* < 0.01, ***P* < 0.001, ****P* < 0.0001, respectively. *P*-values were calculated by two-tailed paired *t* test in most of the experiments unless mentioned otherwise.

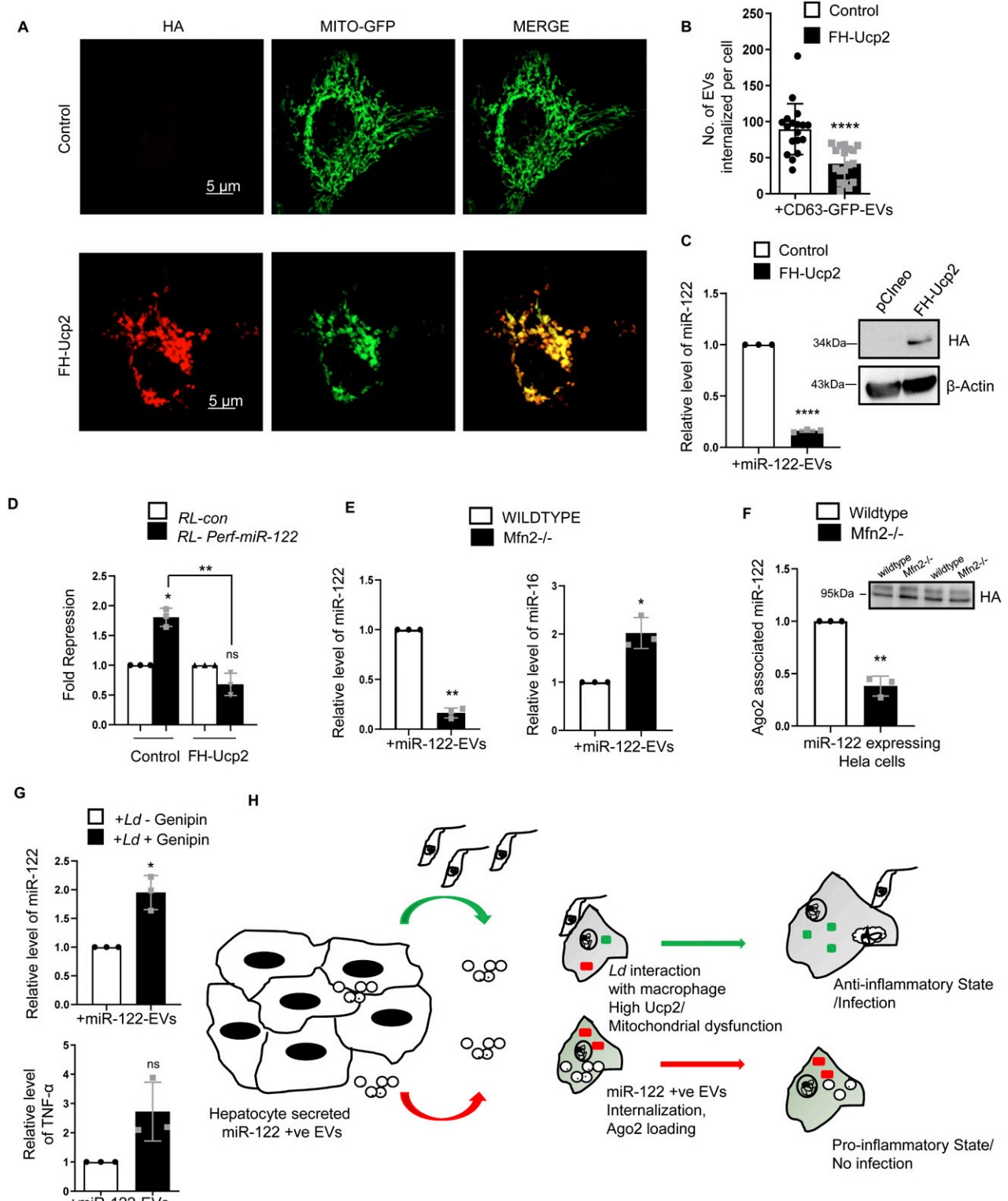

**Figure 3. Mitochondrial depolarization and ER detethering by *Ld* prevents internalization and activity of extracellular vesicle (EV)-derived miRNAs in recipient cells.**
**(A)** Structure of mitochondria (green) in Mito-GFP–expressing HeLa cells transfected with FH-Ucp2 (red) or control plasmid. Scale bar 5 $\mu$m. **(B)** Graphical representation of the internalization of CD63-GFP–positive EVs in recipient HeLa cells transfected either with control or FH-Ucp2–expression plasmids (n = 18, number of cells; P < 0.0001, unpaired *t* test). **(C)** Effect of FH-Ucp2 expression on uptake of miR-122. Relative level of internalized EV-derived miR-122 in recipient HeLa cells transfected either with control or FH-Ucp2 expression plasmids (*left panel*; n = 3 independent experiments; P < 0.0001). U6 was used as endogenous control and values for control plasmid transfected and miR-122 EV-treated cells was considered as unit. *Right panel* shows the Western blot for HA. β-Actin was used as loading control. **(D)** Luciferase assay to

cause the defect in cellular uptake of miRNA-containing EVs in mammalian cells. Mfn2 is a protein responsible for mitochondrial tethering with ER. Mfn2 knockout MEFs showed a decreased ER tethering of mitochondria. To revalidate that the Ucp2 over-expression mediated loss of ER–mitochondria contact is responsible for the reduced EV-associated miRNA internalization and Ago2 association, EV-derived miR-122 levels were compared in Mfn2 knockout and wild-type MEFs. Importantly, as reported earlier, the endogenous miR-16 level was found to be increased in Mfn2 depleted cells (Fig 3E and F) that has been consistent with the previous findings on increased cellular miRNA content upon Ucp2 overexpression (Chakrabarty & Bhattacharyya, 2017). To reconfirm the importance of Ucp2-lowering for cellular entry of EV-associated miRNAs we used genipin, an inhibitor of Ucp2, to see its effect on EV-mediated miR-122 entry. The miR-122 entry and associated increase in TNF-α expression has been observed in infected macrophage cells treated with Ucp2 inhibitor genipin (Fig 3G). Interestingly, ectopic expression of FH-Ucp2 has a marginal effect on low endogenous levels of miR-122 expressed in RAW264.7 cells compared to the changes in miR-122 levels in pmiR-122 transfected RAW264.7 cells (Fig S3D). Conversely, there has been no detectable change in the cellular ATP level or mitochondrial polarization status or Ucp2 expression in RAW264.7 cells ectopically expressing miR-122 from pmiR-122 plasmid (Fig S3E–G). Overall, these data suggest Ld infection–associated increase in Ucp2 that causes mitochondrial detethering with ER and prevents the pro-inflammatory miR-122 entry via EVs into the infected macrophage to stop expression of pro-inflammatory cytokines (Fig 3H).

### Leishmania infection increases cellular and extracellular miR-146a levels in infected cells

The hypothesis that the infected cells should release the EVs enriched with factors to facilitate the propagation of infection inspired us to examine the key protein components that are specifically released by infected cells or their export is prevented in the infection context. The EVs isolated from control and Ld-infected cells were analysed for their protein content. Although upon infection there has been a drop in expression of HRS or Rab27a proteins that are known to have role in late endosome/MVB formation and EV release (Fig 4A and B), we did not detect a significant change in EV diameter or marker protein levels in population isolated from both control and Ld-infected RAW264.7 cells (Fig 4C–E). In the mass spectrometric analysis of released EVs that is followed by candidate protein identification, we have documented several candidate proteins that showed exclusive presence in the EVs isolated from control or Ld-infected cells. However, differential expression of any important regulatory factors of cytokine expression, were not detected (Fig S4 and Tables S1–S3). Therefore, it could be the small RNA population of the infected host that may carry the required anti-inflammatory information to the neighbouring cells as part of EVs. In infected cells, both miR-146a and miR-155, two most important regulatory miRNAs of inflammation pathways, were found to be up-regulated. Interestingly, among them, only miR-146a were detected in EVs released by infected cells and the content of miR-146a had increased several folds in EVs from Ld-infected cells compared with EVs from control noninfected cells (Fig 4F and G). Similar increase in miR-146a in EVs released by infected primary macrophage was also detected (Fig 4H). The miRNAs that are reported to be increased in Ld-infected cells (Chakrabarty & Bhattacharyya, 2017), miR-21, miR-125b, or miR-16 all were found to be increased in the EVs released by infected cells (Fig 4I).

### Leishmania by targeting HuR ensures export of miR-146a but not miR-155 from the infected cells

We report differential export of miR-146a from Ld-infected RAW264.7 macrophage. But how miR-146a gets differentially exported out of infected macrophage cells is a fascinating question. Retention of miRNA and their targets with polysomes has been found to be the reason for reduced miRNA export noted in mammalian cells (Ghosh et al, 2015), whereas target RNA presence positively influences the export process (Ghosh et al, 2021). We have isolated the polysomes from control and infected cells and detected increased retention of both miR-146a and miR-155 with polysomes in the infected cells (Fig S5A and D). The increased retention of IL-10 and MyD88 mRNAs, known to get expressed in infected cells, were also detected more with polysomes isolated from the infected cells but the miR-146a target TRAF6 mRNA and TNF-α mRNA was found to be less with infected cell polysomes (Fig S5B and C).

show the repression levels of RL-perf-miR-122 in recipient cells transfected with pCIneo control or FH-Ucp2–expression plasmids and treated with miR-122–containing EVs. The repression levels were calculated as a ratio of firefly normalized RL-Control value to firefly normalized RL-perf-miR-122 value (n = 3 independent experiments; 0.0117, 0.0987, unpaired t test = 0.0013). **(E)** Relative level of internalized miR-122 (left panel; P = 0.0012) and endogenous miR-16 levels (right panel; P = 0.0316) in Mfn2 (Mitofusin2) wild-type and knockout MEF cells treated with miR-122–containing EVs. Quantification was done by qRT-PCR and U6 levels were used as normalizing control (n = 3 independent experiments). Values for Mfn2 wild-type cells were set as unit. **(F)** FH-Ago2 associated miR-122 levels in Mfn2 wild-type and knockout MEF cells (transfected with FH-Ago2) co-cultured with miR-122 expressing HeLa cells. Immunoprecipitated Ago2 levels were used for normalization of associated miR-122 (n = 3 independent experiments; P = 0.0077). Values for Mfn2 wild-type cells was considered as unit. **(G)** Rescue of EV-miR-122 internalization in Leishmania donovani infected RAW264.7 cells in presence of genipin, the inhibitor of Ucp2. RAW264.7 cells infected with Ld were treated with genipin (100 μM after 4 h of infection) followed by addition of miR-122 positive EVs. Relative levels of miR-122 (top panel; P = 0.0311) and TNF-α mRNA (bottom panel; P = 0.0968) increased in infected RAW264.7 cells treated with genipin (n = 3 independent experiments). U6 levels were used as control for normalization of miRNA and 18s rRNA was used normalizing levels for mRNA. Values for infected and genipin untreated set was considered as unit. **(H)** Diagrammatic representation of the EV-mediated crosstalk between hepatocytes and macrophages. Hepatocytes release miR-122–containing EVs which can be transferred to macrophages that causes production of proinflammatory cytokines and can prevent Ld infection (red arrow). Inversely, Ld-infected macrophages are unable to take up the miR-122–containing EVs due to up-regulation of Ucp2 protein in cells and are tuned to have high production of anti-inflammatory cytokines for sustained parasitic infection (green arrow). Data information: In all experimental data, error bars are represented as mean ± SEM, ns, nonsignificant, *P < 0.05, **P < 0.01, ***P < 0.001, ****P < 0.0001, respectively. P-values were calculated by two-tailed paired t test in most of the experiments unless mentioned otherwise. Positions of molecular weight markers are marked and shown with the respective Western blots.
Source data are available for this figure.

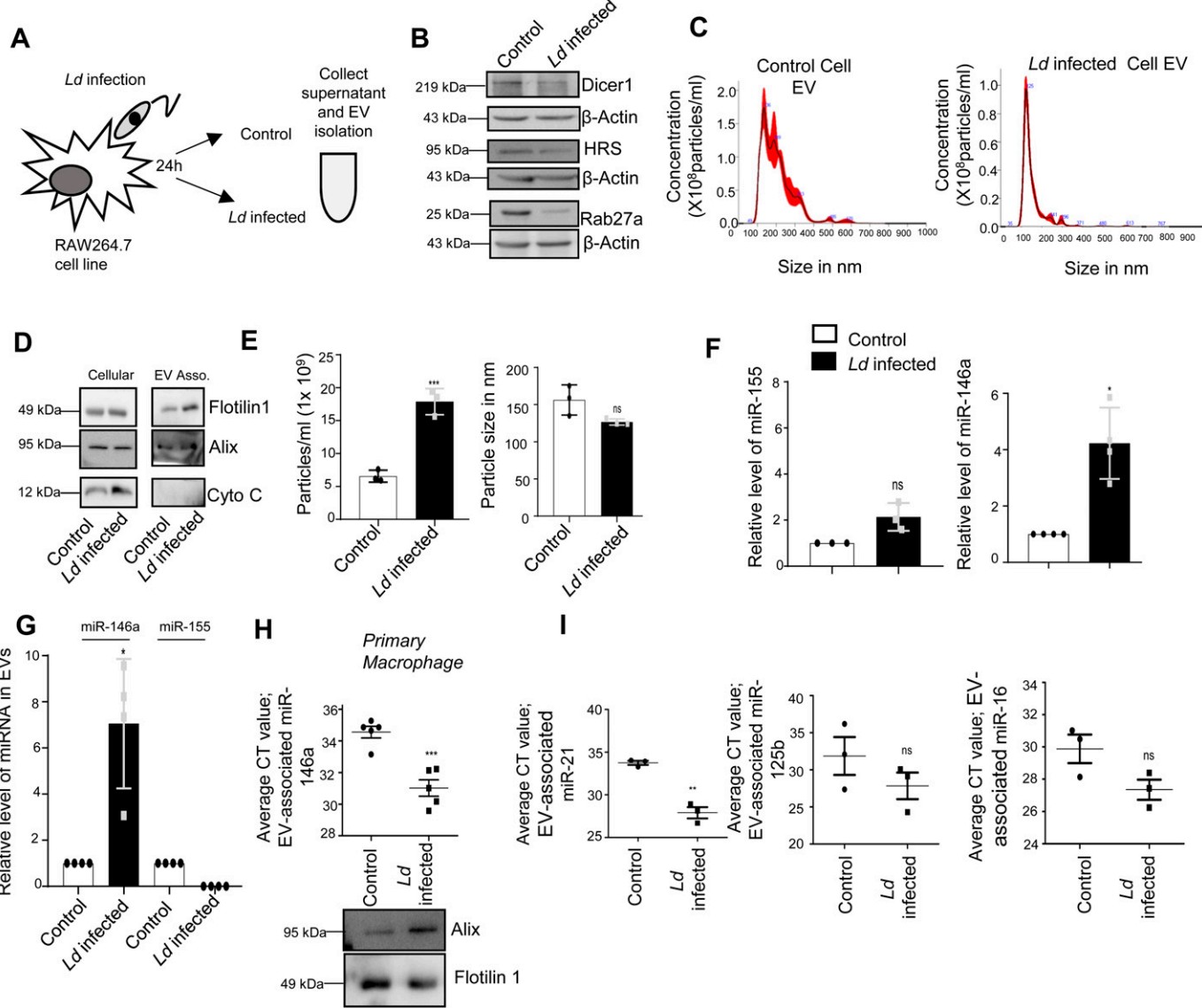

**Figure 4. *Leishmania donovani (Ld)* infection triggers extracellular export of miRNAs from infected macrophage cells.**
**(A)** Schematic representation of the experiment. After 24 h of *Ld* infection, the host macrophage and extracellular vesicles (EVs) isolated from culture supernatant were analysed for cellular and EV-associated protein and RNA analysis. **(B)** *Ld* infection altered cellular protein levels. Western blot images showed protein levels after 24 h of infection of macrophages. β-Actin was used as loading control. **(C, D, E)** Characterisation and quantification of EVs released from control and *Ld*-infected macrophages. Nanoparticle Tracking Analysis (NTA) of EVs released from control (C, *left panel*) and infected cell (C, *right panel*). **(D)** Western blot images showed EV marker proteins in control and infected cell released EVs. **(E)** Number and size of EVs released by control and infected cells were quantified by NTA analysis (E, *left panel*, *P* = 0.0008 and *right panel*, *P* = 0.0677, n = 3 independent experiments, respectively, unpaired *t* test). **(F, G)** miR-146a level increases and the miRNA gets exported from *Ld*-infected macrophage cells. qRT-PCR based relative quantification of cellular miR-155 (F, *left panel*, *P* = 0.0817, n = 3 independent experiments) and miR-146a (F, *right panel*, *P* = 0.0146, n = 4 independent experiments) after 24-h infection of RAW264.7 cells. U6 was used as endogenous control. miRNA levels of infected cells were normalized against noninfected controls. qRT-PCR data represented the relative level of miR-146a and miR-155 in EVs from RAW264.7 cells after infection. EV marker protein Alix was used for normalization of EV-associated miR-146a level (G, *P* = 0.0229, n = 4 independent experiments). **(H)** *Ld* infection triggers the extracellular export of miRNAs from mouse peritoneal macrophages. miR-146a level was also estimated in EV-derived from mouse peritoneal macrophage (H, *upper panel*, *P* = 0.0006, n = 5 independent experiments, unpaired *t* test) and Western blot data also showed EV marker proteins (H, lower panel). **(I)** Average Ct value of miR-21 (I, *left panel*, *P* = 0.0011, n = 3 independent experiments, unpaired *t* test), miR-125b (I, *middle panel*, *P* = 0.2654, n = 3 independent experiments, unpaired *t* test), and miR-16 (I, *right panel*, *P* = 0.0794, n = 3 independent experiments, unpaired *t* test) in EVs released from *Ld*-infected and noninfected RAW264.7 cells. Data information: In all experimental data, error bars are represented as mean ± SEM, ns, nonsignificant, *\*P* < 0.05, *\*\*P* < 0.01, *\*\*\*P* < 0.001, respectively. *P*-values were calculated by two-tailed paired *t* test in most of the experiments unless mentioned otherwise. Positions of molecular weight markers are marked and shown with the respective Western blots. Source data are available for this figure.

The polysome retention data thus could not clearly explain why there has been a preference of miR-146a for export from *Ld*-infected cells. miRNAs accumulate at endosomes before they get packed and exported out (Mukherjee et al, 2016). We had isolated the endosomes and ER fractions from control and infected cells to document preferential enrichment of miR-146a over miR-155 at the endosomal fraction that can explain its enhanced export observed in infected macrophage cells (Fig S5E–G). HuR is known for its role in export of miRNAs in mammalian liver and macrophage cells (Mukherjee et al, 2016). HuR drives miRNA accumulation with endosomes before the export but HuR level is known to get reduced in *Ld*-infected cells (Goswami et al, 2020) (Fig S5H). The HuR level increases after LPS treatment that is associated with enhanced export of miR-155 from LPS-activated macrophage cells (Goswami et al, 2020), it is possible that low HuR in *Ld*-infected cells may cause reduced export of miR-155. Does binding of HuR makes miR-155 available in endosomal fraction and for export? Supporting the notion, we have documented strong interaction of HuR with miR-155 in LPS-treated cells (Fig S5I and J). Similarly export of miR-155 from HA-HuR expressing cells has also been noted (Fig S5K).

## EV-containing miR-146a down-regulates miR-122 maturation in hepatocytes

The infected cells prevent the entry of miR-122–containing EVs thereby restricting the activation of the *Ld*-infected cells by miR-122 to prevent the death of the internalized parasites owing to enhanced proinflammatory response observed in miR-122 recipient macrophages (Chen et al, 2011). However, miR-122 in hepatic EVs could also get transferred to a naïve neighbouring macrophage not infected with *Ld*. How the parasite prevents the activation of resident neighbouring naïve macrophages to prevent the proinflammatory cytokine accumulation in the infection microenvironment? It is likely that *Ld* must have adopted a mechanism to reduce the miR-122–containing EV release by neighbouring hepatocytes in the infection niche to prevent miR-122–mediated activation of naïve noninfected macrophage cells. Can *Ld* do it through cross talk with hepatocytes via the EVs released by infected cells? We have noted miR-146a as the predominant anti-inflammatory miRNA in the EVs secreted by the infected macrophage (Fu et al, 2017). Does miR-146a reduce the miR-122 level in the hepatocytes?

We have isolated the EVs from infected macrophage for treating human hepatoma cell Huh7. We have documented an increase in miR-146a content of recipient hepatic cells with a reduction in mature miR-122 level there (Fig 5A and B). The decrease of mature miR-122 was accompanied by an increase in pre-miR-122 level and with an increase in endogenous control miR-16 level in the hepatocyte. Thus, the miR-146a–mediated lowering of hepatic miR-122 is a miRNA specific effect. To confirm that the effect of miR-146a on mature miR-122 level is miR-146a specific, we have isolated the EV from RAW264.7 cells ectopically expressing miR-146a and used the isolated EVs for the treatment of the hepatocytes (Fig 5C). The EVs isolated from control and miR-146a expression plasmid transfected cells were analysed by NTA to document no major change in size and number of the EVs under miR-146a overexpression condition (Fig S6A). Increased miR-146a content in EVs released from pmiR-146a–expressing cells was also noted (Fig S6B) and cellular and EV

fractions were Western blotted for EVs and cytosolic markers to confirm the purity of the fractions (Fig S6C). The miR-146a–containing EV treatment enhanced cellular miR-146a level in recipient hepatocytes with concomitant decrease in mature miR-122 and increase in pre-miR-122 levels observed there (Fig 5D). To substantiate the idea of miR-146a–induced reduction in miR-122 levels in hepatocytes, we have expressed miR-146a in hepatocytes and documented decreased miR-122 level associated with increased pre–miR-122 and miR-146a levels in Huh7 cells (Fig 5E). We have noted a reduction in Dicer1 level upon *Ld*-infected cell EV treatment of hepatocytes. However, miR-146a–containing EV treatment or miR-146a expression in hepatocytes, did not alter Dicer1 level (Fig 5F–H). Thus, altered pre–miR-122 processing by Dicer1 in miR-146a enriched hepatocyte could not be explained by the unchanged Dicer1 level. The increased pre–miR-122 however got associated with the Dicer1 to a lesser extent in Huh7 cells expressing pmiR-146a where total pre–miR-122 level increases (Fig 5J and K). Therefore, there must be a mechanism that excludes the pre–miR-122 association with Dicer1 for its subsequent processing to the mature form. The miR-146a expression, however, enhances the P-ERK1/2 level and a decrease in P-p38 level (Fig 5I), suggesting a possible lowering of transcriptional events for pre-miR-122 known to be linked with P-p38 level (Basu & Bhattacharyya, 2014).

## Infected cell secreted EVs downgrade inflammatory response in neighbouring macrophage and induce macrophage polarization

Although lowering of miR-122 levels in hepatocytes could be one of the mechanisms to reduce the proinflammatory response in the naïve macrophage, there must be additional mechanisms to restrict the production of inflammatory cytokines in the neighbouring naïve macrophage cells adjacent to the infected macrophages. To score the effect of infected cell EVs on naïve macrophage cells, we treated naïve RAW264.7 cells with EVs isolated from *Ld*-infected RAW264.7 cells (Fig 6A). Interestingly, when the naïve RAW264.7 cells were pretreated with *Ld*-infected cell–derived EVs, we documented a lesser responsiveness and relatively low proinflammatory cytokine production in the treated cells when exposed to pro-inflammatory agents such as LPS. Remarkably, the microvesicles (MVs) isolated from the infected cells increase the production of cytokines in recipient cells on LPS exposure (Fig 6B).

To score the effect of infected cell EVs on polarization of naïve macrophage, we treated the RAW264.7 macrophage with *Ld*-infected EVs (Fig 6C) and measured the amount of internalized miRNA and found an increase in miR-146a content. This was accompanied by a reduction in iNOS expression (Fig 6D). With EV treatment, we noted no change in Dicer1 level and a decrease in HuR expression with increased levels of P-p38 and P-ERK1/2 levels (Fig 6E). These were accompanied by an increase in IL-10 expression and decrease in expression of proinflammatory cytokine IL-1β (Fig 6F). To conclude on casualty of EV-associated miR-146a with changed cytokine expression observed in *Ld*-infected cell–derived EV-treated cells, we used EVs released by pmiR-146a–expressing RAW264.7 cells and treated naïve RAW264.7 cells with miR-146a–containing EVs. We noted an increase in IL-10 and a decrease in IL-1β as well as iNOS mRNAs after miR-146a–containing EV treatment (Fig 6G and H). Increased expression of miR-146a has also found to be associated

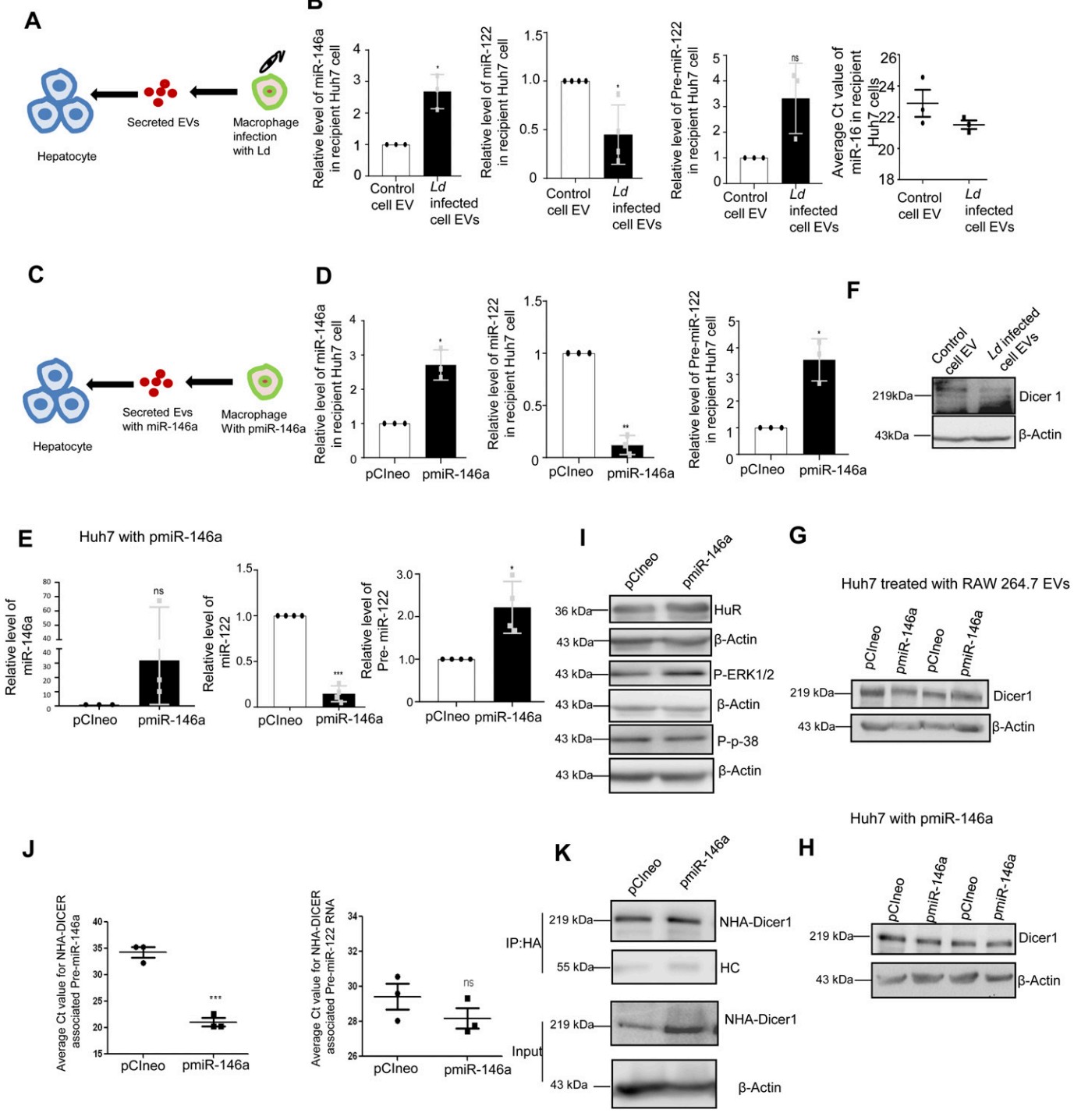

**Figure 5. Extracellular vesicle (EV)–mediated transfer of *Ld*-infected cell derived miR-146a reduces miR-122 level in hepatic cells.**
**(A, B)** Effect of control and *Ld*-infected cell–derived EVs on naïve hepatocytes. **(A)** A schematic representation of the experiment (A). **(B)** Relative levels of miR-146a (B, *left panel*, *P* = 0.0329, n = 3 independent experiments), miR-122 (B, *middle left panel*, *P* = 0.0364, n = 4 independent experiments), Pre-miR-122 (B, *middle right panel*, *P* = 0.0993, n = 3 independent experiments), miR-16 (B, *right panel*, unpaired *t* test *P* = 0.2074, n = 3 independent experiments) were measured by qRT-PCR in recipient hepatic Huh7 cells after 24 h of control and infected cell–derived EV treatment. U6 was used as endogenous control for miRNA normalization and for Pre-miR-122 level, GAPDH was used as endogenous control. Values for control EV-treated cells were used for normalization of infected cell–EV-treated cells. **(C, D)** Effect of pmiR-146a–expressing macrophage derived-EV treatment on recipient hepatocytes. **(C)** A schematic representation of the experiment has been described in panel. **(D)** Relative levels of miR-146a (D, *left panel*, *P* = 0.0213, n = 3 independent experiments), miR-122 (D, *middle panel*, *P* = 0.0036, n = 3 independent experiments) and Pre-miR-122 (D, *right panel*, *P* = 0.0307, n = 3 independent experiments) were measured by qRT-PCR in recipient Huh7 cells after pCIneo and pmiR-146a transfected RAW264.7 cell–derived EV treatment for 24 h. U6 was used as endogenous control for cellular miRNA level normalization. Values for pCIneo-transfected cell–derived EV-treated cell were set as unit. For Pre-miR-122 level, GAPDH was used as endogenous control. **(E)** Effect of pmiR-146a overexpression on miR-122 level in hepatocytes. Relative levels of miR-146a (E, *left panel*, *P* = 0.2251, n = 3 independent experiments), miR-122 (E, *middle panel*, *P* = 0.0003, n = 4

with decreased iNOS mRNA and NO levels in RAW264.7 cells (Fig 1C) associated with a decrease in TNF-α protein level (Fig 1C) and cellular IL-1β protein expression (Fig S6D), respectively. We expressed miR-146a in naïve and *Ld*-infected RAW264.7 cells and have noted a decrease in IL-1β and TNF-α expression and increase in IL-10 expression (Fig 6I). Interestingly, inhibition of miR-146a by anti-miR-146a reduced the expression of IL-10 both in uninfected and *Ld*-infected cell–derived EV-treated RAW264.7 cells, whereas the activation of miR-146a–expressing cells with LPS have a induced effect on anti-inflammatory cytokine IL-10 mRNA level (Fig 6J and K). This data suggests an M2 polarization of macrophage expressing miR-146a and that become refractory to immuno-stimulation by LPS (Fig 6I–K). To further reaffirm the contribution of *Ld*-derived factors associated with EVs-released by infected cells on naïve macrophage cell polarization and IL-10 production, we treated RAW264.7 cells with *Ld*-derived EVs and lipophosphogylcan (LPG) isolated from the *Ld* membrane. We have noted no important change in the expression of major signalling component in *Ld*-derived EV-treated cells (Fig S7A, B, and F). As reported earlier, we had noted decreased TNF-α expression in both sets of treated cells, whereas IL-1β only got reduced in presence of *Ld*-derived EVs but not with *Ld* LPG (Fig S7C and E, [de Carvalho et al, 2019]). The IL-10 level decreased in both *Ld*-derived EVs or LPG treatment and therefore, these Leishmania-derived factors cannot be the major player for the observed increase in IL-10 in neighbouring macrophage cells (Fig S7C and G). iNOS mRNA level, however, decreases with *Ld*-derived EV treatment (Fig S7B), whereas the *Ld*-derived EV, LPG, or soluble antigens of Leishmania (SLA) has an insignificant effect on endogenous level of miR-122 in RAW264.7 cells (Fig S7D, E, and G). On contrary, the effect of *Ld*-derived EVs, LPG, or SLA also has an inhibitory effect on endogenous miR-146a levels in hepatocytes as observed in Huh7 cells (Fig S7D, E, and G). Taken together, the data suggest that the effect of miR-146a rather than the *Ld*-derived factors, as part of infected cell secreted EVs, has the potential to change the IL-10 level in recipient naïve macrophage cells and is essential for macrophage M2 polarization. Thus, we conclude that the EV-associated miR-146a of *Ld*-infected cell EVs, have an anti-inflammatory effect on naïve macrophage cells and is associated with an increase in IL-10 production in EV-treated cells, whereas the LPS responsiveness as well as pro-inflammatory cytokine expression level remain reduced in miR-146a–enriched RAW264.7 cells.

### HuR is required for miR-146a–mediated IL-10 induction

From the data discussed so far, it has become more likely that miR-146a control the IL-10 level in macrophage cells. How does miR-146a

ensure a high IL-10 expression in treated macrophage to polarize them to have an anti-inflammatory response pathway—a prelude to the infection niche establishment? HuR is an important post-transcriptional gene regulator in mammalian macrophage and is essential for activation and expression of proinflammatory cytokines in macrophage exposed to LPS. Activated macrophage also showed increased expression of HuR with LPS treatment time (Goswami et al, 2020). Interestingly, expression of HA-HuR also enhances the expression of pro-inflammatory cytokines in untreated macrophages and could get the cells to the activated state (Fig 7A and B, Goswami et al, 2020). Therefore, to have the strong anti-inflammatory effect of miR-146a, the HuR-mediated proinflammatory effect should be neutralized or countered by the transferred miR-146a from the infected cells. *Leishmania* could down-regulate HuR in mammalian macrophage cells (Fig S5H). However, it is possible that the extracellular miR-146a released by the infected cells could bring down the effect of HuR in noninfected cells present in the infection niche to ensure an overall anti-inflammatory pathway prevalent in all immune cells present there. miR-146a–mediated down-regulation of HuR has been reported previously (Cheng et al, 2013). Does miR-146a counter the action of the HuR to stop inflammatory responses in macrophage cells? In miR-146a–containing EV-treated cells, we have noted that the expression of HuR could not enhance the IL-1β or TNF-α expression in RAW264.7 cells. Surprisingly, an increase in IL-10 expression in HuR expressing cells was detected when treated with miR-146a–containing EVs (Fig 7C and D). Inversely, siHuR treatment causes a reduction in IL-10 expression in cells treated with miR-146a–containing EVs (Fig 7E and F). However, miR-146a mimic decreases HuR levels as well as HuR induced increase in TNF-α and IL-1β expression, whereas IL-10 level increases in presence of HA-HuR when miR-146a is also expressed in RAW264.7 cells (Fig 7G–K). How does HuR ensure the high IL-10 level? It is known that miR-21 negatively regulates IL-10 (Wang et al, 2017) and HuR is also known to interact and sponge out miR-21 to inactivate it in mammalian cells (Poria et al, 2016). We have noted reduction in miR-21 level in HA-HuR–expressing cells, whereas siHuR treatment prevents miR-21 lowering. We have also noted increased miR-21 export in HuR-expressing cells that suggests HuR-mediated export of miR-21 that helps the IL-10 expression (Fig 7L and M). Thus, HuR by up-regulating IL-10 level also ensures the strong anti-inflammatory polarization in miR-146a–containing EV-treated cells. Interestingly, HuR eventually gets decreased in cells expressing high levels of miR-146a to possibly balance the HuR-mediated stabilization of cytokine mRNAs such as TNF-α and IL-1β possibly to restrict excess

independent experiments) and Pre-miR-122 (E, *right panel*, P = 0.0279, n = 4 independent experiments) were measured by qRT-PCR in Huh7 cells after pCIneo and pmiR-146a expression in hepatocytes. U6 was used as endogenous control for miRNA normalization. pCIneo transfected cells was used for normalization of pmiR-146a transfected cells. For Pre-miR-122 level, GAPDH was used as endogenous control. **(F, G, H)** Effect of EV treatment on Dicer1 level in naïve hepatocytes. **(F)** Level of Dicer1 was measured by Western blot in control and *Ld*-infected cell–EV-treated Huh7 cells. β-Actin was used as loading control (F). **(G)** Dicer1 level was also measured in Huh7 treated with EVs derived from pCIneo and pmiR-146a transfected RAW264.7 cells. β-actin was used as loading control. **(H)** Effect of pmiR-146a expression on Dicer1 in Huh7 hepatocytes was also determined. β-Actin was used as loading control. **(I)** Effect of pmiR-146a overexpression on signalling component proteins in hepatocytes. Levels of HuR, P-ERK1/2, P-p38 were determined by Western blot analysis in pCIneo and pmiR-146a transfected Huh7 cells. β-Actin was used as loading control. **(J, K)** Effect of NHA-Dicer1 expression on Dicer1 associated precursor miRNA level in cells expressing pmiR-146a. **(J)** Average Ct value of NHA-Dicer1 associated Pre-miR-146a (J, *left panel*, P = 0.0005, n = 3 independent experiments, unpaired *t* test) and Pre-miR-122 level (J, *right panel*, P = 0.2630, n = 3 independent experiments, unpaired *t* test) were determined by Real-time PCR after HA-Dicer1 was immunoprecipitated from pCIneo and pmiR-146a transfected Huh7. **(K)** NHA-Dicer1 was immunoprecipitated using anti-HA antibody and was detected in immunoprecipitated and input samples using anti-HA antibody. HC, heavy chain. Data information: In all experimental data, error bars are represented as mean ± SEM, ns, nonsignificant, *P < 0.05, **P < 0.01, ***P < 0.001, respectively. P-values were calculated by two-tailed paired *t* test in most of the experiments unless mentioned otherwise. Positions of molecular weight markers are marked and shown with the respective Western blots. Source data are available for this figure.

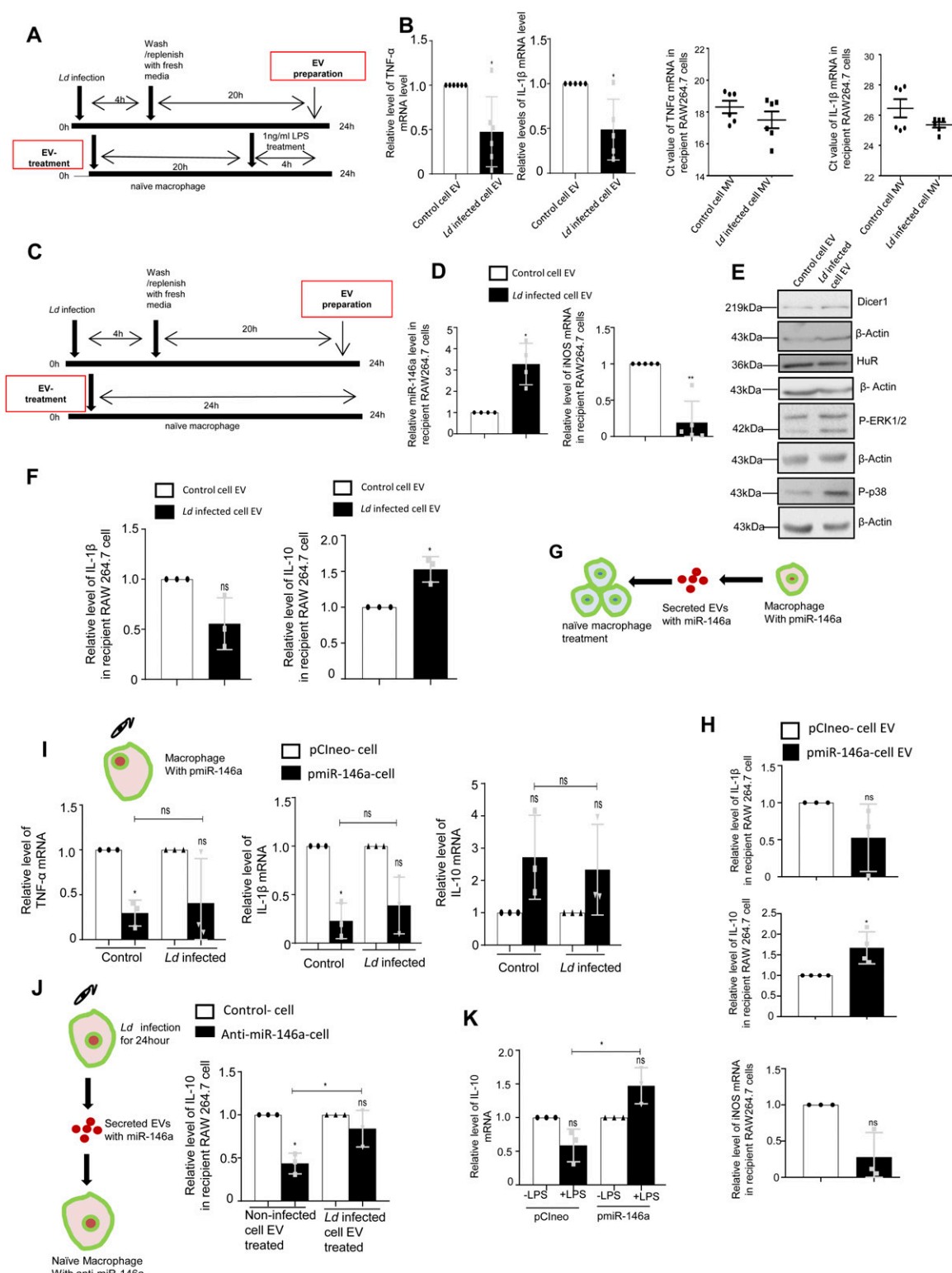

**Figure 6.    Extracellular vesicle (EV)–mediated transfer of miR-146a from infected macrophage cells promotes anti-inflammatory response in naïve recipient macrophages.**
**(A, B)** Effect of EV treatment on LPS induced activation of recipient macrophages. **(A)** A schematic representation of the experiment has been shown in the left. **(B)** Relative cellular levels of TNF-α (B, *left panel*, P = 0.0224, n = 6 independent experiments), IL-1β (B, *middle left panel*, P = 0.0280, n = 5 independent experiments) mRNAs were measured by qRT-PCR from Infected cell–EV-treated macrophages after LPS treatment. TNF-α and IL-1β (B, *middle right*, P = 0.2487, n = 6 independent experiments, unpaired *t* test and *right panel*, P = 0.1133, n = 6 independent experiments, unpaired *t* test, respectively) were measured by qRT-PCR from infected cell derived Microvesicle (MV)–treated macrophages followed by 4 h of LPS (1 ng/ml) treatment. 18s rRNA was used as endogenous control. Values in the noninfected cell EV treated set were used as units.

anti-inflammatory response in miR-146a expressing cells. Therefore, it is the balanced expression of HuR and miR-146a that determine the polarization status of the macrophage. In miR-146a compromised situation, expression of HuR promotes M1 polarization (Goswami et al, 2020), while HuR-mediated miR-21 export contributes in increased IL-10 expression. High IL-10, in the presence of miR-146a, possibly down-regulates HuR protein level to ensure the subdued expression of proinflammatory cytokines and expression of IL-10 itself as a feedback effect (Rajasingh et al, 2006).

### miR-146a acts as a balancer of IL-10 production and *Ld* infection

Does miR-146a uptake via EV affect the Ld infection process? We have measured the number of parasites internalized in the treated macrophage after *Ld*-infected cell EVs or miR-146a–containing EVs treatment. We have performed quantification and noted reduced entry of *Ld* after the miR-146a positive or infected cell EV treatment (Fig 8A–D). Lower expression level of *Leishmania* amastigote–specific gene amastin also confirmed reduced levels of *Ld* entry after the miR-146a–containing EV or infected cell EV treatment of RAW264.7 cells (Fig 8E–G). miR-146a expression in RAW264.7 cells also reduces the *Ld* infection process. We have used single cell infection level examination to conclude on preferential reduction of *Ld* internalization in cells expressing miR-146a (Fig 8H). Infection level may be down-regulated due to a problem in signalling pathways or downstream factors. We have noted an increased P-p38 level in miR-146a–expressing macrophage that signifies the existence of a possible counteractive pathway that usually gets reduced in cells infected with *Ld* where a p38/MAPK down-regulation has been noted (Fig 8M). The decreased *Ld* infection could have also been attributed by a possible reduction in the endocytosis process itself. However, in a latex bead internalization assay, we did not document any reduction in latex bead internalization after miR-146a–containing EV or infected cell EV treatment of RAW264.7 cells (Fig 8I–L).

## Discussion

EVs are known for their role in intercellular communication. They play an important role in immune response during any pathological condition and also help to establish tumour microenvironment (Thery et al, 2002). In disease condition, EVs either can help in progression of the disease or in providing protection against the disease. There are many studies where it has been reported that exosomes from *Mycobacterium* infected cells can trigger proinflammatory response in noninfected cells (Bhatnagar et al, 2007). *Leishmania* parasites release exosomes as a vehicle to transport proteins to the host cell for immunosuppressive action (Silverman et al, 2010). *Leishmania* protein GP63 was found to be transported to neighbouring hepatocytes via secreted exosomes that target Dicer1, a processor of pre-miRNAs, to down-regulate expression of liver specific miR-122 (Ghosh et al, 2013). *Leishmania* parasite secretes exosomes within sandfly midgut and these exosomes are part of the sandfly inocula during bite which helps in pathogenesis via overproduction of IL-17a in target cells (Atayde et al, 2015). Gioseffi et al (2020), in recent time, have reported *Leishmania*-infected EVs as a carrier of large number of Leishmanial and host proteins (Gioseffi et al, 2020). Among the proteins exported out, they have identified that a *Leishmania* homolog of mammalian angiogenesis promoting factor, vasohibin is found to induce endothelial cells to release angiogenesis promoting factors and thereby helping in promotion of lesion vascularization during infection.

miRNAs, being the regulator of expression of several cytokines, are important players in determining the fate of immune cells and in particular have a major role in buffering the immune activation processes by regulating the expression of several cytokines directly or indirectly (Lindsay, 2008; Testa et al, 2017). miR-146a and miR-155 are the two most important players that are explored for their potential role as immune checkpoint regulators in the mammalian system. Whereas miR-155 is a pro-inflammatory miRNA, miR-146a

---

**(C, D, E, F)** Effect of infected cell–derived EV treatment on recipient macrophages. **(C)** A schematic representation of the experiment has been shown (C). **(D)** Relative levels of miR-146a and iNOS mRNA in recipient macrophage after 24 h of EV treatment were measured by qRT-PCR and miRNA and mRNA levels in noninfected cell EV-treated cell were set as unit. U6 and GAPDH was used as endogenous control for miRNA and mRNA, respectively (D, *left panel* P = 0.0184, n = 4 independent experiments and *right panel*, P = 0.0035, n = 5 independent experiments, respectively). **(E)** Levels of Dicer1, HuR, P-ERK1/2, and P-p38 were measured by Western blot analysis in infected cell–derived EV-treated RAW264.7 cells. β-Actin was used as loading control. **(F)** Relative levels of IL-1β (F, *left panel*, P = 0.0967, n = 3 independent experiments) and IL-10 (F, *right panel*, P = 0.0357, n = 3 independent experiments) were measured by qRT-PCR in recipient RAW264.7 cells after 24 h of EV treatment. GAPDH was used as endogenous control and values for noninfected cell EV-treated cells were set as unit. **(G, H)** Effect of EV-mediated transfer of miR-146a on recipient macrophage. **(G)** A schematic diagram has been given. **(H)** Relative levels of cytokine mRNAs; IL-1β, IL-10 and iNOS (H, *upper panel*, P = 0.2131, n = 3 independent experiments, H, *middle panel*, P = 0.0406, n = 4 independent experiments and H, *lower panel*, P = 0.0662, n = 3 independent experiments, respectively). GAPDH was used as endogenous control and values for pCIneo control EV-treated cells were set as units. **(I)** Effect of *Ld* infection on cytokine mRNA levels in pmiR-146a overexpressed macrophages. A schematic diagram of the experiment has been shown (I, upper panel). Relative cytokine mRNA level of TNF-α (I, *left panel*, P = 0.0138, P = 0.1740, P = 0.7287, n = 3 independent experiments), IL-1β (I, *middle panel*, P = 0.0183, P = 0.0684, P = 0.4727, n = 3 independent experiments), and IL-10 (I, *right panel*, P = 0.1491, P = 0.2417, P = 0.7457, n = 3 independent experiments) were measured by qRT-PCR from pCIneo and pmiR-146a transfected RAW264.7 cells after 24 h of *Ld* infection or no infection. GAPDH was used as endogenous control and values for pCIneo control cells were set as units (I, unpaired *t* test was performed for comparison between pmiR-146a–expressing noninfected control and infected set). **(J)** Effect of control and infected cell derived-EV treatment on recipient macrophages transfected with control or miR-146a inhibitor oligos. A schematic diagram has been given (J, left panel). Relative mRNA level of IL-10 was measured by qRT-PCR from miR-146a inhibitor transfected and EV-treated RAW264.7 cells (J, right panel, paired *t* test, P = 0.0149,0.3208, unpaired *t* test was performed for comparison between miR-146a inhibitor transfected control EV and infected cell EV-treated sets, P = 0.0456, n = 3 independent experiments, respectively). GAPDH was used as endogenous control and values for negative control inhibitor transfected cells were set as units. **(K)** Effect of LPS treatment on pmiR-146a–expressing macrophages. Relative level of IL-10 was measured by qRT-PCR from control or pmiR-146a–expressing cells with or without LPS exposure (1 ng/ml) (K. paired *t* test, P = 0.0984, 0.0921, unpaired *t* test was performed for comparison between LPS-treated sets, P = 0.0132, n = 3 independent experiments). GAPDH was used as endogenous control and values for untreated control cells were set as units. Data information: In all experimental data, error bars are represented as mean ± SEM, ns, nonsignificant, *P < 0.05, **P < 0.01, respectively. P-values were calculated by two-tailed paired *t* test in most of the experiments unless mentioned otherwise. Positions of molecular weight markers are marked and shown with the respective Western blots.

Source data are available for this figure.

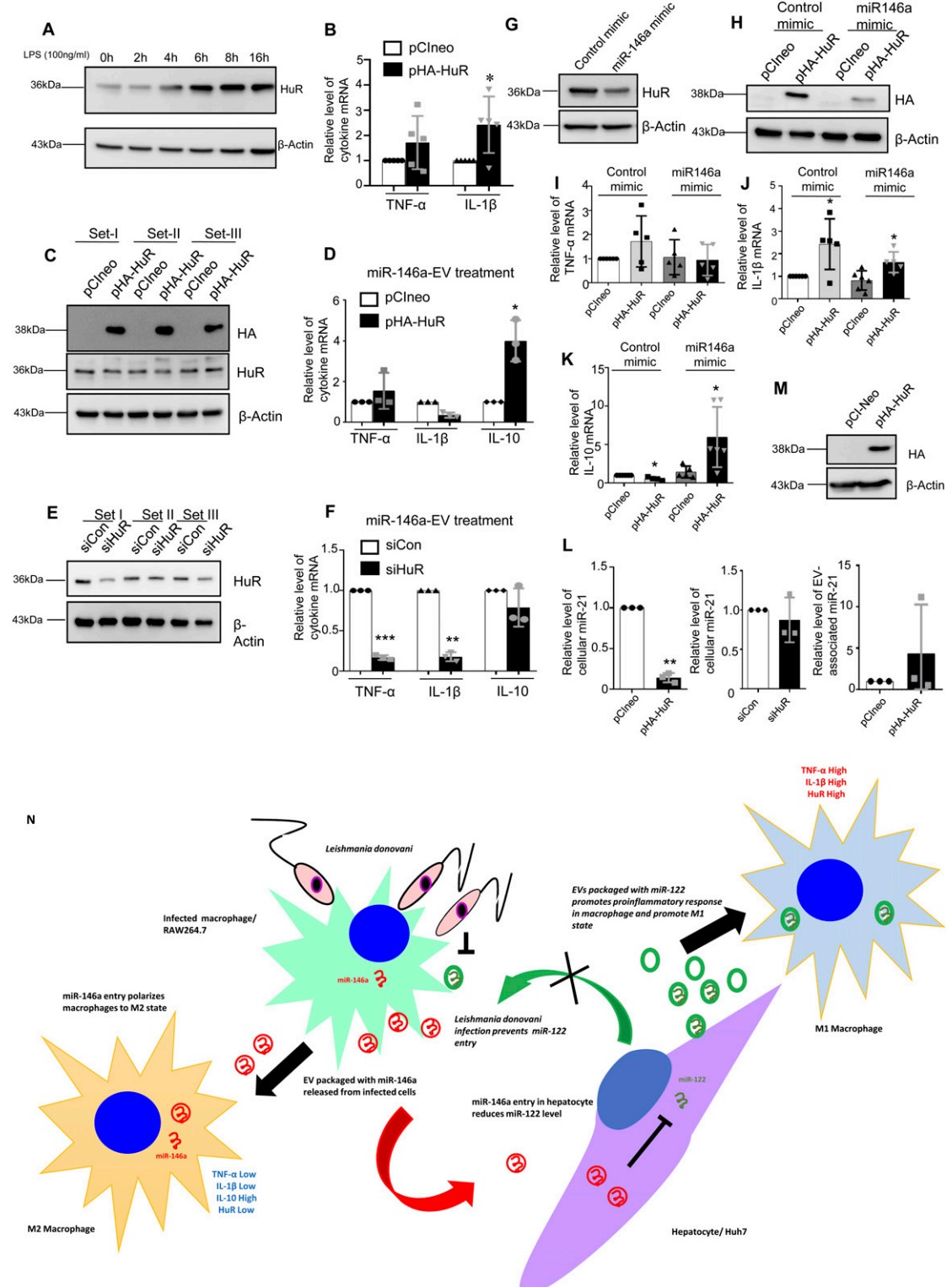

**Figure 7. Crosstalk between miRNA-exporter HuR and miR-146a determines the fate of cytokine response in macrophage cells.**
**(A)** LPS treatment (100 ng/ml) for different time points in RAW264.7 cells. The cell lysates were immunoblotted for HuR. β-Actin was used as loading control. **(B)** Real-time PCR was performed for proinflammatory cytokine mRNAs (TNF-α and IL-1β) in pCIneo and pHA-HuR cells co-transfected with pre-miR control mimic. GAPDH was used for normalization (n = 5 independent experiments; P-values = 0.2048, 0.0467, respectively). Values observed in pCIneo transfected cells were considered as units for normalization of values obtained with pHA-HuR transfected cells. **(C, D)** Extracellular vesicles (EVs) isolated from pmiR-146a transfected RAW264.7 cells were treated to recipient RAW264.7 cells expressing pCIneo or pHA-HuR. **(C)** Recipient cells treated with pmiR-146a–expressing cell EVs were immunoblotted for HA and HuR. β-Actin was

acts oppositely to balance the immune activation process (Testa et al, 2017). In the infection context, the pathogens need to target the miRNA pathways to control the inflammation level. The *Ld* parasite has shown previously to control expression of miRNAs in a negative manner in infected cells. By down-regulating HRS and depolarizing mitochondria in the host, *Ld* affects the miRNA recycling process to enhance the miRNA content of the infected cells. However, the accumulated miRNAs are largely non-functional as they were found to be accumulated with endosomal fraction and they fail to recycle for new round of target repression on ER attached polysomes (Chakrabarty & Bhattacharyya, 2017; Bose et al, 2020). The miR-146a that also increases with *Ld* infection gets exported to neighbouring cells. The miR-122 is another miRNA in hepatocytes that *Ld* needs to control to ensure lower cholesterol biogenesis in the liver cells essential for the parasite survival. *Ld*, by targeting Dicer1 in hepatocytes, control the expression of miR-122 in infected liver tissue (Ghosh et al, 2013; Chakrabarty & Bhattacharyya, 2017). Our results show that miR-146a, transferred from infected macrophage cells, also has a negative role to play on miR-122 expression in mouse liver. Reciprocally, miR-122 can get transferred from activated or stressed liver cells to resident macrophage to get them activated (Ghosh et al, 2015; Mukherjee et al, 2016). The parasite uses a secondary mechanism of lowering the miR-122 in liver cells to ensure reduced transmission of the proinflammatory miR-122 to macrophages to prevent the expression of inflammatory cytokines there. EV-mediated crosstalk between the macrophages and hepatocytes fosters an environment of cell-to-cell communication to take place. However, in the liver, it is known that the EVs released from LPS-treated THP-1 macrophages could stimulate the proliferation of hepatic stellate cells, whereras miR-103-3p present in the EVs isolated from macrophages could get transferred to the hepatic stellate cells to promote its growth (Chen et al, 2020).

Mitochondrial activity is found to be a determining factor for intercellular transfer of miRNAs. Does miR-122 affect mitochondrial function? miR-122 is known to regulate mitochondrial metabolic function. Cationic amino acid transporter gene CAT-1 level decreases, whereras PPARGC1A (PGC-1α) and succinate dehydrogenase subunit A and B level increases in HCC cells expressing miR-122. PGC-1α is the regulator of mitochondrial biogenesis and it is also involved in energy metabolism. Succinate dehydrogenase (SDH) is the enzyme complex located in inner mitochondrial membrane and is involved in both TCA cycle and electron transport chain. PGC-1α and succinate dehydrogenase both are found to be the putative secondary targets of miR-122 (Burchard et al, 2010). Mitochondrial depolarization as well as ATP content are found to be non-responsive to miR-122 levels in macrophage cells.

How mitochondria affects the EV entry in the recipient cells? We have found how the mitochondria-ER tethering and mitochondria–endosome interaction play a key role in miRNA internalization process which also affect miRNA turnover in mammalian cells (Chakrabarty & Bhattacharyya, 2017). Any factor that can influence these interactions will also modify organellar dynamics in mammalian cells to eventually affect the EV-internalization. Increased expression of Ucp2 due to *Ld* infection or exogenous expression of FH-Ucp2 is associated with mitochondrial depolarization and changed mitochondrial dynamics that resulted in defective miRNA compartmentalization. Importantly Mfn2 loss in *Ld*-infected cells or Mfn2 null condition in Mfn2−/− MEF cells are also associated with miRNA compartmentalization defect (Chakrabarty & Bhattacharyya, 2017). Therefore, microRNA compartmentalization or defective EV uptake possibly directly gets affected by the change in mitochondrial dynamics rather than changed membrane potential or ATP level. *p*-Triflouromethoxyphenylhydrazone (FCCP) is an uncoupler of mitochondrial membrane potential but does not affect cellular ATP content whereas oligomycin inhibit F0F1-ATPase

---

used as loading control. **(D)** Real-time PCR was performed for cytokine mRNAs (TNF-α, IL-1β, and IL-10) in pCIneo and pHA-HuR transfected recipient cells treated with pmiR-146a–EVs. GAPDH was used for normalization (n = 3 independent experiments; *P*-values: 0.3952, 0.0134, and 0.0352, respectively). Values obtained with pCIneo transfected and EV-treated cells were considered as units. **(E, F)** EVs were isolated from pmiR-146a transfected RAW264.7 cells and used to treat the recipient RAW264.7 cells depleted of HuR (siHuR transfected). **(E)** Recipient cells treated with pmiR-146a–EVs were immunoblotted for HuR to check for proper silencing and β-Actin was used as loading control (E). **(F)** Real-time PCR was performed for cytokine mRNAs (TNF-α, IL-1β and IL-10) in siCon or siHuR-containing recipient cells treated with pmiR-146a–EVs, GAPDH was used for normalization (n = 3 independent experiments; *P*-values: 0.0004, 0.0015, 0.2641, respectively) (F). In this particular Experiment, siCon transfected and EV-treated cells was used for normalization of values noted in siHuR-containing cells. **(G, H, I, J, K)** Effect of pHA-HuR expression in RAW264.7 cells transfected with pre-miR control mimic or pre-miR-146a mimic. Transfection was performed either with pCIneo (control plasmid) or pHA-HuR. **(G)** Immunoblotting of the cell lysates for HuR using β-Actin as loading control (G). **(H)** Immunoblotting of the respective cell lysates for HA to check for proper overexpression of HA-HuR, β-Actin was used as loading control. Real-time PCR was performed for TNF-α, where GAPDH has served as control. **(I)** Values obtained for pCIneo and pre-miR control mimic co-transfected sets were taken as units (n ≥ 5 independent experiments; *P*-values: 0.2048, 0.8587, 0.8620, respectively). **(J)** Real-time PCR was performed for IL-1β, GAPDH served as control. Values obtained for pCIneo and pre-miR control mimic co-transfected sets were taken as units (n ≥ 5 independent experiments; *P*-values: 0.0467, 0.3453, 0.0375, respectively). **(K)** Real-time PCR was performed for IL-10, where GAPDH has served as control. Values obtained for pCIneo and pre-miR control mimic co-transfected sets were taken as units (n ≥ 4 independent experiments; *P*-values: 0.0239, 0.2710, 0.0267, respectively). **(L, M)** Relative level of miR-21 in HA-HuR expressing RAW264.7 cells. Real-time PCR for miR-21 level in RAW264.7 cells expressing pHA-HuR and transfected with pre-miR-146a mimic. Values obtained for pCIneo and pre-miR mimic co-transfected sets were taken as units. U6 snRNA served as control (n = 3 independent experiments; *P*-value = 0.0015) (L, left panel). Real-time PCR for miR-21 level in recipient RAW264.7 cells depleted of HuR which were treated with miR-146a–containing EVs. Values obtained for siCon and miR-146a–containing EV-treated sets were taken as units. U6 snRNA served as control (n = 3 independent experiments; *P*-value = 0.5259) (L, middle panel). Real-time PCR of EV-associated miR-21 level from EVs of RAW264.7 cells expressing or not expressing pHA-HuR. EV marker protein Flotillin-1 was used for normalization of miR-21 level (n = 3 independent experiments; *P*-value = 0.4313) (L, right panel). Values obtained for pCIneo transfected cell–derived EVs was considered as unit. **(M)** Immunoblotting of the cell lysates with HA using β-Actin as loading control in pCIneo or pHA-HuR transfected RAW264.7 cells. **(N)** The proposed model depicts the role of EV-associated miRNAs in *Ld* infection. The left half shows that *Ld* infection triggers EV-mediated secretion of miR-146a from macrophage which when transfers to naïve macrophage polarizes it toward M2 phenotype because of its anti-inflammatory role whereas the right half represents what happens when miR-146a get delivered to naïve hepatocyte via EV. miR-146a reduces miR-122 level in hepatocytes. If not restricted, miR-22 as part of hepatocyte secreted EVs polarizes naive macrophage to M1 phenotype because of its proinflammatory role. Data information: In all experimental data, error bars are represented as mean ± SEM, *P < 0.05; **P < 0.01; ***P < 0.001. *P*-values were calculated by two-tailed paired *t* test in most of the experiments unless mentioned otherwise. Positions of molecular weight markers are marked and shown with the respective Western blots. Source data are available for this figure.

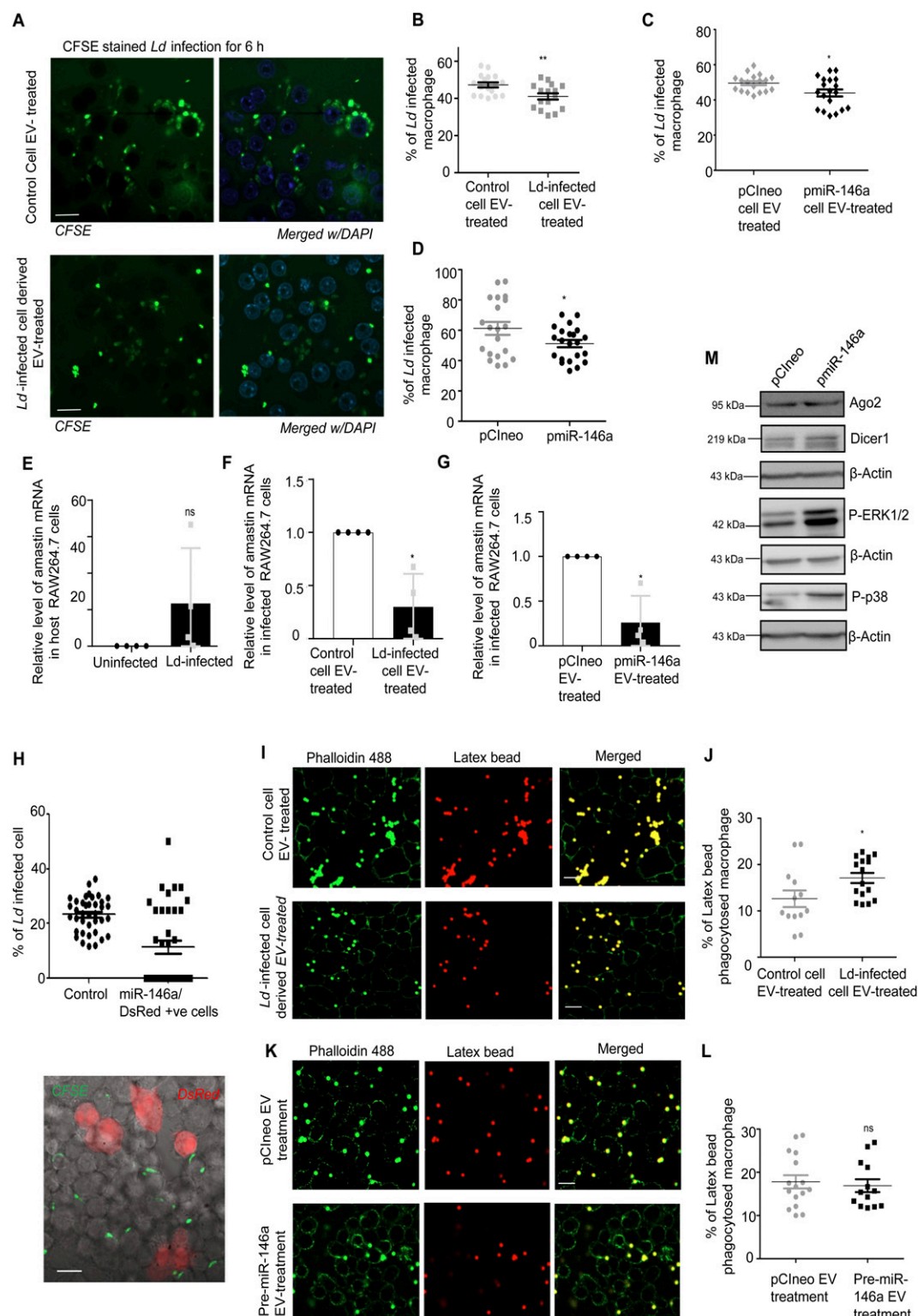

**Figure 8. Effect of extracellular vesicle (EV)–associated miR-146a treatment on *Ld* infection.**
**(A, B, C, D)** Infected cell–derived EV treatment decreases *Ld* infection in recipient cells. **(A)** Parasites were labelled with CFSE dye and infection was given for 6 h after 24 h of control and infected cell–derived EV treatment (A, *upper* and *lower panel*, respectively). Parasite infection was determined by counting the CFSE positive structures inside cells. Scale bar 10 μm. **(B)** Percentage of infected cells was calculated for both control cell EV and infected cell EV-treated RAW264.7 cells (B, P = 0.0064, unpaired *t* test, n ≥ 16 number of fields). **(C)** Parasites were labelled with CFSE dye and infection was given for 6 h after 24 h of pCIneo and pmiR-146a–transfected cell–derived EV treatment. Percentage of infected cells was calculated for both pCIneo and pmiR-146a transfected cell–derived EV-treated cells (C, P = 0.0208, unpaired *t* test, n ≥ 18

and prevents ATP production without affecting mitochondrial membrane potential (Chakrabarty & Bhattacharyya, 2017). We found oligomycin treatment could not inhibit EV-mediated internalization of miR-122 and there was no change in miR-122 uptake in 3-h FCCP treated cells. Thus, it is not the mitochondrial depolarization or ATP level, but is the change in mitochondrial dynamics and its interaction with endosomes or ER in Mfn2 KO or FH-Ucp2–expressing or *Ld*-infected cells (Chakrabarty & Bhattacharyya, 2017) that probably hampers endocytosis of miR-122–containing EVs in recipient cells.

MicroRNAs are the small noncoding RNAs that are involved in post-transcriptional and translational regulation of various genes. Thus, altered expression of these miRNAs is one of the main reasons for various pathogenic conditions (Bartel, 2018). Earlier reports suggested up-regulation of miR-146a/b during *Salmonella* infection (Ordas et al, 2013). Up-regulation of miR-155 and miR-146a/b and down-regulation of miR-20a, miR-191, and miR-378 in *Mycobacterium avium*–infected condition were also reported (O'Connell et al, 2012; Curtale et al, 2019). miR-155 favours the pro-inflammatory environment, whereas miR-146a favours the anti-inflammatory environment (Lindsay, 2008). Recently miR-146a–mediated M2 polarization during *L. donovani (Ld)* infection has been reported. The investigation revealed that *L. donovani (Ld)* infection up-regulates miR-146a expression which favoured parasite survivability in hosts by maintaining the anti-inflammatory environment. Inhibition of miR-146a lowered parasite burden in infected Balb/C mice as well as anti-inflammatory environment (M2 phenotype). Interestingly, this up-regulation of miR-146a during *Ld* infection is regulated by Super Enhancer components like BET bromodomain 4 (BRD4) and P300 (Das et al, 2021).

The polarization of macrophage is contributed by several factors including specific activation of signalling component that favours either the TLR4-p38/MAPK-NF-kB–mediated activation of proinflammatory cytokine expression or it may be an inhibitory circuit involving ERK1/2–dependent inactivation of the proinflammatory cytokines with concomitant increase in expression of anti-inflammatory cytokine like IL-10 (Mukherjee et al, 2021). The parasite seems to use the infected cell EVs packed with miR-146a to transmit the anti-inflammatory signals to the naive neighbouring macrophage to polarize them to the M2 stage. It is not known whether the polarization is reversible, but subsequent treatment of miR-146a overexpressed cells with LPS suggest the refractory nature of the recipient macrophage to inflammatory response. Thus, miR-146a–containing EVs could be considered as a good immune modulator and use of that could be important to control the inflammation in different physiological contexts like in prevention of viral infection related cytokine storm (Arisan et al, 2020) or inflammation associated with tumour (Rupaimoole et al, 2016). The cross communication of miR-146a–containing EVs to other noninfected tissue is another important aspect. Do EVs with miR-146a get transferred via the bloodstream to the spleen and do they have any effect on subsequent establishment of infection and T-cell polarization observed in the spleen of infected animals? These are important questions for future exploration.

In our experimental system, the effect of Leishmanial secretory proteins on the M2 state of macrophages seems to be marginal as the cytokine pattern of recipient macrophages treated with *Ld* promastigote culture–derived EVs showed decreased level of IL-10 and IL-1$\beta$ as well. Thus, these results clearly explain that the observed M2 state of the naïve macrophages treated with infected cell–derived EVs is due to EV-mediated transfer of miR-146a and possibly not by *Leishmania* specific factors. The results obtained from the SLA or LPG or parasite-derived EV-treated macrophages and hepatocytes suggest *Leishmania* derived factors are not primarily responsible also for miR-122 and miR-146a up-regulation in respective cell types (Fig 7N).

Although miR-122 can up-regulate IL-1$\beta$ mRNA, we could not detect an increase in mature IL-1$\beta$ by ELISA in miR-122 expressing cells. Thus, the transcriptional surge of IL-1$\beta$ mRNA due to miR-122 expression is not associated with increase detectable in the protein level. It is possible that the excess IL-1$\beta$ mRNA produced in the presence of miR-122 may get transferred to other immune cells where they may elicit a proinflammatory response. Otherwise, the inflammatory response observed in miR-122 expressing cells is manifested primarily by TNF-$\alpha$ rather than IL-1$\beta$.

HuR is a RNA binding protein with immense functional diversity. HuR is known to stabilize the mRNAs with AU-rich elements in their 3′ UTR and thus have a role in the inflammation process as most of the proinflammatory cytokines such as TNF-$\alpha$ or IL-1$\beta$ bear the AU-rich elements in their 3′ UTR (Srikantan & Gorospe, 2012). In this report, we have detected a non-canonical role of HuR in stabilization of IL-10 mRNA. The anti-inflammatory IL-10 mRNA is regulated negatively by miR-21 (De Melo et al, 2021), the suppression of

number of fields). **(D)** Parasites were labelled with CFSE dye and infection was given for 24 h to pCIneo and pmiR-146a–transfected cell. Percentage of infected cells was calculated for both pCIneo and pmiR-146a transfected cells (*P* = 0.0401, unpaired *t* test, n ≥ 16 number of fields). **(E)** RAW264.7 cells were infected with *Ld* for 24 h and infection was measured in terms of amastin mRNA level by qRT-PCR (E, *P* = 0.2213, n = 4 independent experiments). **(F)** Relative level of amastin mRNA was determined by qRT-PCR from control and infected cell–derived EV-treated cells followed by 6 h infection (F, *P* = 0.0205, n = 4 independent experiments). **(G)** Relative level of amastin mRNA from pCIneo and pmiR-146a transfected cells after 24 h of infection was determined by qRT-PCR (G, *P* = 0.0160, n = 4 independent experiments). GAPDH was used as endogenous control. Values for uninfected cell or control EV-treated cell or pCIneo transfected cells were considered as units. **(H)** RAW264.7 cells were co-transfected with DsRed and pmiR-146a and was infected with *Ld* for 24 h and infection level was visualized at individual cells (lower panel) and quantified (upper panel). Percent of infection in pmiR-146a/DsRed expressing versus non-transfected *Ld*-infected cells were measured. Scale bar 10 *μ*m. (n = 38, Number of fields). **(I, J, K, L)** Latex bead phagocytosis after EV treatment. Red fluorescent bead (Fluorescent Red; Sigma-Aldrich) was diluted in medium and added to EV-treated RAW264.7 cells at 1:10 (cell:bead) ratio for 6 h. Red fluorescent bead was visualized by a confocal microscope. **(I, J)** Phalloidin 488 was used for staining the cytoskeleton of cells (I). Scale bar 10 *μ*m. Percentage of beads phagocytosed in EV-treated macrophages was plotted and compared between control cell EV-treated and *Ld*-infected cell EV-treated cells (J, unpaired *t* test, *P* = 0.0347, Number of fields, n ≥ 13 number of fields). **(K, L)** In panel (K) visualization of beads phagocytosed in EV-treated macrophages was monitored in pCIneo transfected control cell EV-treated and pmiR-146a–expressing cell EV-treated cells. The quantification was completed and plotted in panel L (unpaired *t* test, *P* = 0.6761, Number of fields, n ≥ 13 number of fields). Scale bar = 10 *μ*m. **(M)** Cellular levels of the major signalling pathway components in pmiR-146a and pCIneo expression conditions were determined by Western blot. *β*-Actin was used as loading control. Data information: In all experimental data, error bars are represented as mean ± SEM, ns, nonsignificant, *, ** represent *P*-value of <0.05, <0.01, respectively. *P*-values were calculated by two-tailed paired *t* test in most of the experiments unless mentioned otherwise. Positions of molecular weight markers are marked and shown with the respective Western blots.

miR-21 activity by HuR has been reported previously (Poria et al, 2016). In the context of miR-146a induced expression of IL-10, HuR acts synergistically to remove miR-21 from the cell to boost the anti-inflammatory IL-10 production in macrophages. Interestingly, in the context of an anti-inflammatory environment the miRNA antagonistic role of HuR predominate to ensure expression of miRNA-repressed cytokine mRNAs, whereas in the proinflammatory context, the mRNA's stabilizer role of HuR is important to ensure proinflammatory cytokine expression there.

# Materials and Methods

### Cell culture, peritoneal macrophage isolation, and parasite infections

Human hepatic Huh7, mouse hepatic HePa1-6, and Mfn2 wild-type and knockout MEFs cells (WT/Mfn2$^{-/-}$) were cultured in Dulbecco's Modified Eagle's Medium (Gibco-BRL) and 10% FBS (heat-inactivated foetal bovine serum). In case of RAW264.7 macrophage cells, RPMI-1640 medium (Gibco) with 2 mM L-glutamine and 0.5% $\beta$-mercaptoethanol along with 10% FBS was used. Primary murine peritoneal macrophages were isolated from BALB/c mice subjected to intraperitoneal injection of 1 ml of 4% starch solution. Cold 1× PBS was used to lavage or wash the peritoneal cavity to isolate the peritoneal macrophages the following day which were then pelleted and plated. These cells were also cultured in RPMI-1640. EV isolation of Peritoneal Exudate Cells (PECs) was done from a 60 mm plate. All experiments with PEC were carried out after 48 h of isolation (Goswami et al, 2020). For macrophage or Huh7 activation, LPS dose of 1 or 100 or 500 ng/ml was used for different time points depending on the experiments. RAW264.7 and Huh7 cells were stimulated with LPS from *Escherichia coli* O111:B4 (Calbiochem). For EV-free growth medium, 10% EV-depleted FBS (made by ultracentrifugation of FBS at 1,000,00$g$ for 4 h) was added to DMEM or RPMI-1640 for respective cell types.

### Animal experiments

BALB/c male mice (4–6 wk) were divided into four groups (three animals each) of either uninfected or infected and treated or untreated with EVs. Two groups (six animals) were infected with 1 × 10$^7$ promastigotes by intracardiac puncture and maintained for 30 d. On the 30$^{th}$ day, EVs were injected into six animals (three infected and three uninfected with parasites), whereas 1× PBS was injected in the remaining six animals (three infected and three uninfected). For EV isolation, HePa1-6 cells were transfected with miR-122–expressing plasmid and ~5 × 10$^9$ EVs (measured by Nanoparticle Tracker Nanosight NS300) isolated from these transfected cells were suspended in 1× PBS and injected into the tail vein. The animals were euthanized on the 31$^{st}$ day for Kupffer cell isolation.

All the experiments were carried out in accordance with the National Regulatory Guidelines issued by the Committee for the Purpose of Supervision of Experiments on Animals, Ministry of Environment and Forest, Govt. of India. The animal experiments were performed in agreement with the Institutional Animal Ethics Committee. The BALB/c mice were kept under controlled conditions (temperature 23 ± 2°C, 12 h/12 h light/dark cycle) in individually ventilated cages.

Kupffer cell isolation was done from control and experimental group of mice. The animals were anaesthetized, and livers were perfused with HBSS via the hepatic portal vein with an incision in the inferior vena cava until they turned bloodless and then digested with a Liver Digest Medium. Livers were then excised, minced, and filtered through a 70-$\mu$m cell strainer. The resultant single cell suspension was centrifuged at 50$g$ for 5 min to pellet and store the hepatocytes. The supernatant was then collected and loaded onto a Percoll gradient of 25% and 50%. It was centrifuged at 1,200$g$ for 30 min at 4°C without brakes. The interphase containing the Kupffer cells was collected and washed twice with PBS and the pellet was stored. The Kupffer cell pellet and hepatocyte pellet were subjected to RNA isolation by TriZol reagent.

For detection of purity of isolation, the Kupffer cells of all the four groups were compared with the respective hepatocytes by quantification of mRNA levels of C-type Lectin Domain Family 4, Member F (Clec4f), a Kupffer Cell marker, and hepatocyte marker, Albumin. The kinetoplast DNA levels were also quantified in the four groups to detect the level of leishmanial infection.

### Parasite culture and infection to macrophage cell line

*L. donovani (Ld)* strain AG83 (MAOM/IN/83/AG83) was obtained from a Indian Kala-azar patient and was maintained in golden hamsters (Chakrabarty & Bhattacharyya, 2017). Amastigotes were obtained from infected hamster spleen and transformed into promastigotes. Promastigotes were maintained in M199 medium (Gibco) supplemented with 10% FBS (Gibco) and 1% Pen-Strep (Gibco) in 22°C. RAW264.7 cells or PEC were infected with stationary phase *Ld* promastigotes of second to fourth passage at a ratio of 1:10 (cell: *Ld*) for 6, 16, or 24 h depending upon the experiment (Goswami et al, 2020).

### Plasmid constructs and transfection

pCIneo, precursor miR-146a–overexpressing plasmid (pmiR-146a), precursor miR-122–overexpressing plasmid (pmiR-122), and HA-HuR plasmid(pHA-HuR) were transfected using Fugene HD (Promega) for RAW264.7 cells as described previously (Goswami et al, 2020). For Huh7 cells, all transfections of plasmids were performed using Lipofectamine 2000 reagent (Life Technologies) according to the manufacturer's protocol. For miRNA inhibitor (anti-miR-146a) experiments, 30 pmol per well (30 nM) was transfected using RNA imax reagent in RAW264.7 cells according to manufacturer's protocol. Transfection of Negative control inhibitor was used as controls. For co-transfection of plasmid and pre-miR mimic, Lipofectamine 2000 (Invitrogen) was used. 50 pmoles of pre-miR mimic were transfected per well of a 12 well plate (50 nM). siRNA transfection was performed using RNAi max (Invitrogen) following the manufacturer's instructions. RAW264.7 cells were transfected with 50 pmoles of siRNA per well of a 12-well plate (50 nM). siRL was used as siCon for the experiments. EV treatment was given to the recipient cells 48 h after siRNA transfection and the cells were harvested 24 h after

EV treatment. Donor RAW264.7 cells were transfected with 2 μg pmiR-146a plasmid per well of a six-well plate using Fugene HD.

### EV isolation and characterization

RAW264.7 cells were plated in a 90 mm plate and given infection with stationary phase parasite at about 80% confluency and kept for 6 h. After 6 h of infection, the medium was discarded and cells were washed with PBS to remove extracellular unattached parasites and cells were replenished with fresh RPMI-1640 medium supplemented with 10% EV depleted FBS and 1% Pen-Strep for 20 h. Total infection time was 24 h before EV isolation. After 24 h, the supernatant was collected for EV isolation. Briefly, the supernatant first centrifuged at 3,000$g$ for 15 min at 4°C followed by 30 min centrifugation at 10,000$g$ at 4°C. For all EV isolation experiments, supernatant was passed through a 0.22-$\mu$m filter unit followed by ultracentrifugation on a 30% sucrose cushion at 1,00,000$g$ for 70 min. The supernatant was discarded leaving a medium layer of interface-containing EVs behind, which was then washed with 1× PBS at 1,00,000$g$ for 70 min again. The pellet then was resuspended in PBS for NTA or 1× passive lysis buffer (PLB) for RNA or protein precipitation (Promega). For EV-associated RNA and protein, the pellet was resuspended in PLB. One-third was used for total RNA isolation using TriZzol LS reagent (Invitrogen) following the manufacturer's protocol. Two-thirds of the PLB sample was used for protein precipitation using methanol precipitation. In brief, for 200 $\mu$l of PLB sample, 640 $\mu$l of sterile water, 480 $\mu$l of methanol (Merck), and 160 $\mu$l of chloroform (Sigma-Aldrich) were added and vortexed well followed by centrifugation at 20,000$g$ for 5 min at room temperature. Then the supernatant was discarded and 300 $\mu$l of methanol was added. The mixture was vortexed well and again centrifuged for 30 min at 20,000$g$ 4°C. The supernatant was discarded. The pellet was air dried for 10 min and then dissolved in 1× SDS Dye and heated at 95°C before run on a SDS–PAGE.

For experiments with EV treatment, culture supernatant first centrifuged at 3,000$g$ for 15 min at 4°C followed by 30 min centrifugation at 10,000$g$ and filtered pass through 0.22 $\mu$m following ultracentrifugation at 1,00,000$g$ for 70 min. The pellet was then resuspended in RPMI-1640 medium supplemented with 10% EV depleted FBS and 1% Pen-strep for further use.

For nanoparticle tracking analysis (NTA), the control and infected RAW264.7 cell–derived EV pellet was resuspended in 300 $\mu$l of sterile PBS and resuspended well. Then, diluted 10-folds in PBS and 1 ml of diluted sample was used for NTA (Nanosight NS300).

### EV treatment of recipient cells

After EV isolation, the pellet was resuspended in fresh RPMI-1640 supplemented with 10% EV–depleted FBS and 1% Pen-Strep followed by filtration through a 0.22-$\mu$m filter unit. The EVs were then added to the recipient macrophage or hepatic cell line for 24 h. Post-LPS treatment was given at a dose of 1 ng/ml for 4 h after 20 h of EV treatment to macrophages. For treatment, EVs isolated from 8 × 10^6 cells (one 90 mm plate) were added to 2.4 × 10^5 recipient cells (per well of a 12 well plate). For estimation of parasite infection after EV treatment, parasites were added to EV-treated cells for 6 h.

### Optiprep density gradient ultracentrifugation

For cell fractionation using 3–30% iodixanol gradient (Optiprep gradient) 1.6 × 10^7 cells were used as described earlier (Mukherjee et al, 2016). Optiprep (Sigma-Aldrich) solution was used to prepare 3–30% gradient. In brief, the cell pellet was homogenized in a buffer with glass Dounce homogenizer. The lysate was centrifuged at 1,000$g$ twice and the supernatant was loaded on the gradient and centrifuged at 36,000 rpm in a Beckman-Coulter SW60Ti rotor for 5 h. Approximately 400 $\mu$l of 10 fractions were collected for RNA and protein.

### Total RNA isolation and quantification of miRNA and mRNA

Total RNA extract from experimental samples using TriZol reagent (Invitrogen) for cell and TriZol LS (Invitrogen) for Optiprep/subcellular fraction according to manufacturer's protocol. All miRNA and mRNA quantifications were performed as described previously (Basu & Bhattacharyya, 2014; Mukherjee et al, 2016). In brief, miRNA estimation was performed with 100 ng of RNA for cellular and EV samples using specific primers for human miR-146a, mouse miR-155, human miR-122, human miR-21, human miR-16, miR-125b, and U6 snRNA was used as endogenous control. For the assay, one-third of the reverse transcription mix was subjected to PCR amplification with TaqMan Universal Master Mix No AmpErase (Applied Biosystem) and respective TaqMan assay reagents for target miRNAs. Samples were analysed in triplicates for each biological replicates. The comparative $C_t$ method which involved normalization by U6 snRNA was used for quantification (Mukherjee et al, 2016).

For mRNA, 200 ng of total RNA was used for estimation based on SYBR Green based real-time cDNA estimation using specific primer for target genes. The comparative $C_t$ method which involved normalization by GAPDH or 18s rRNA was used for relative quantification of mRNA (Goswami et al, 2020). The details of Primers, miRNA assays are provided as part of Tables S4 and S5. Information on expression plasmids, siRNAs, miRNA-mimic and miRNA inhibitors are part of Table S6.

### Western blot

Western blot analyses were performed as described elsewhere (Mazumder et al, 2013). Imaging of the blots was performed using an UVP BioImager 600 system equipped with VisionWorksLife Science software (UVP) V6.80. ImageJ software was used for quantification. Details of all antibodies used for Western blot and immunofluorescence experiments are summarized in Table S7.

### Immunofluorescence

For internalization, *Ld* parasites were stained with 1 $\mu$M carboxyfluorescein succinimidyl ester (CFSE) dye (green) for 30 min at 22°C followed by PBS wash thrice and then resuspended in medium before adding to the cell. Cells were fixed with 4% PFA for 20 min after three PBS wash. Nuclei were stained with DAPI. For calculating the percentage of infected cells, a minimum number of 100 cells were counted. Cells were observed under the Zeiss LSM800 confocal microscope.

## Polysome isolation

For polysome isolation ~8 × 10[6] cells were used as starting material. After 24 h of infection, cells were scraped in PBS and pelleted down at 600$g$ for 5 min at 4°C. The cell pellet was collected and incubated in 1× lysis buffer (10 mM Hepes, 25 mM KCL, 5 mM MgCl$_2$, 1 mM DTT, 5 mM VRC, 1× PMSF, cycloheximide 100 $\mu$g/ml, 1% Triton X-100, and 1% sodium deoxycholate) for 15 min at 4°C followed by centrifugation at 3,000$g$ for 10 min at 4°C. Then the supernatant was collected and centrifuged at 20,000$g$ for 10 min at 4°C. The supernatant was collected and loaded on 30% sucrose cushion in gradient buffer and centrifuged at 31,200 rpm for 1 h in 4°C in a SW61Ti rotor. After centrifugation the non-polysome fractions were collected and the rest of the solution was mixed followed by centrifugation at 31,200 rpm for 30 min at 4°C in a SW61Ti rotor. The pelleted polysome was suspended in polysome buffer (10 mM Hepes, 25 mM KCL, 5 mM MgCl$_2$, 1 mM DTT, 5 mM VRC, and 1× PMSF) and kept for RNA isolation and Western blot (Ghoshal et al, 2021).

## LPG extraction and treatment

LPG isolation was performed as mentioned earlier (Goswami et al, 2020). In brief, 2 × 10[8] stationary phase parasites (AG83) were used for LPG extraction. The pellet was resuspended in 2 ml of Chloroform: Methanol: Water mixture (1:2:0.5 vol/vol) followed by vortexing and sonication for 10 s thrice at 30 s interval and incubated in mutarotator at room temperature for 2 h. Then the lysate was centrifuged at 4,000$g$ for 30 min at 4°C. The pellet was used for extraction in 9% 1-Butanol in water. After vortex and sonication as before, pellet was incubated for 1 h at room temperature in mutarotator. Second extraction was done with the pellet in 9% 1-Butanol in water again after centrifugation at 4,000$g$ for 30 min at 4°C. The supernatants of both extractions were pooled for lyophilization. The lyophilized LPG was resuspended in 1 ml sterile PBS before cell culture treatment. For treatment, 1:50 (cell: LPG) dose was used for different time points.

## SLA preparation

SLA was prepared from the stationary-phase promastigotes (~10[8] cells) as described earlier (Basu et al, 2005). Briefly, stationary phase parasites were pelleted down at 3,000$g$ for 10 min followed by repeated cycles of freezing (–70°C) and thawing (37°C) followed by 5 min incubation on ice. These cells were then further completely lysed by sonication thrice for 30 s and centrifuged for 10,000$g$ for 30 min at 4°C. The supernatant was used for protein estimation using the Bradford assay. Macrophages or Huh7 were treated with 10 $\mu$g/ml SLA for 24 h.

## Mass spectrometry

For proteomics sample preparation, exosomes from two 90 mm were pooled and protein was precipitated with 100% chilled acetone in –20°C overnight. The protein precipitated was then washed with 80% chilled acetone followed by another wash with 40% chilled acetone. The pellet was then air dried for 5–10 min to remove excess acetone at room temperature and resuspended in 50 mM ammonium bicarbonate (AmBic) followed by sonication thrice. 85 mM DTT was then added, and the sample was heated at 60°C for 1 h. Sample was then incubated in dark with 55 mM Iodoacetamide (IAA) at RT for 30 min. Trypsin (1 $\mu$g/$\mu$l) digestion was carried out to the sample for overnight at 37°C. The reaction was stopped with 100% formic acid (5% of total volume) and snap frozen until lyophilization. Lyophilized sample was used for proteomics and run in Orbitrap analyzer.

## Latex bead phagocytosis assay

Red fluorescent labelled Latex bead (Fluorescent Red; Sigma-Aldrich) was diluted in RPMI-1640 medium and added to macrophages at a ratio of 1:10 (cell:bead) for 6 h. Cells were fixed with 4% PFA as described earlier and mount in DAPI. Cell cytoskeleton was stained with Phalloidin 488. Red latex bead phagocytosis was imaged in Zeiss LSM800 confocal microscope.

## Cytokine measurement by ELISA

Protein level of TNF-$\alpha$ was measured from culture supernatant of RAW264.7 macrophages using sandwich ELISA Kit (BD Pharmingen) as per the manufacturer's protocol. Protein level of IL-1$\beta$ was measured from RAW264.7 cell lysate (1 mg/ml) instead of cell culture supernatant using sandwich ELISA Kit (BD Pharmingen) as per the manufacturer's protocol. Cytokine level were determined by measuring the OD at 450 nm using a microtiter plate reader.

## Measurements of NO in cell culture supernatant

Nitric oxide (NO) was measured from RAW264.7 cell supernatant using Greiss reagent. In brief, equal volume of sample was incubated with equal volume of 1% sulphanilamide for 15 min in dark at room temperature followed by equal volume of 0.1% NEDD (N-1-napththylethylenediamine dihydrochloride) was added and incubated until colour developed. NO level was determined by measuring the OD at 550 nm using a microtiter plate reader.

## Flow cytometry analysis for mitochondrial depolarization

Approximately 0.4 × 10[6] RAW264.7 cells were scraped and washed in PBS. Then the cells were incubated in 100 nM MitoTracker Red for 30 min in 37° C followed by washing with PBS to remove excess stain and finally analysed in BD FACS Fortessa.

## Cellular ATP-level measurement

Amount of ATP was determined using the ATP bioluminescent assay Kit (Sigma-Aldrich). Approximately 1.5 × 10[6] RAW264.7 cells were scraped and washed in PBS. Briefly, the cells were lysed and the equal volume of lysate was added to equal volume of ATP assay mix solution followed by rapid mixing and the amount of light produced was determined by luminometer.

### Immunoprecipitation

Immunoprecipitation was performed as described earlier with minor modifications (Goswami et al, 2020). Briefly, protein G agarose beads (Life Technologies) was used for exogeneous HA protein or any other endogenous protein. Beads were washed twice with 1× IP buffer (20 mM TRIS–HCL, pH 7.5, 150 mM KCL, 5 mM MgCl$_2$, and 1 mM DTT 1× EDTA free protease inhibitor cocktail) at 2,000$g$ for 2 min at 4°C. Beads were then blocked with 5% BSA in lysis buffer for 1 h at 4°C in mutarotator followed by washing with IP buffer as mentioned previously. After blocking, required amount of antibody was added to the bead for another 4 h incubation at 4°C before lysate addition. The final dilution of antibody was 1:100. Cells were lysed in a lysis buffer (1× IP buffer with 0.5% Triton X-100 and 0.5% of sodium deoxycholate) for 30 min in 4°C and sonicated thrice for 10 s interval and 30 s incubation in ice. Then lysate was collected after centrifugation at 16,000$g$ for 10 min at 4°C and was added to the bead-antibody mixture for overnight immunoprecipitation to occur at 4°C. After the reaction, the lysate–bead–antibody mixture was washed with 1× IP buffer thrice and was resuspended in 400 $\mu$l IP buffer. Equal volume was kept for protein and RNA. For RNA, TriZol LS was added thrice of the volume and for protein 5× SDS dye was added so that the final concentration would be 1×.

### Statistical analysis

All graphs and statistical significance were calculated using GraphPad Prism 5.00 and 8.00 version (GraphPad). Experiments were carried out for minimum three times to get $P$-value, unless otherwise mentioned. Two-tailed paired and unpaired $t$ tests were used for determination of $P$-values. The sample size was chosen by convenience and no exclusion criteria were used.

### Ethics statement

Balb/C mice and Syrian golden hamsters were obtained from CSIR-Indian Institute of Chemical Biology animal house facility. All experiments were performed according to the national regulatory guidelines stated by the Committee for the Purpose of Supervision of Experiments on Animals, Ministry of Environment and Forest, Govt. of India. All animal experiments were performed with prior approval of the Institutional Animal Ethics Committee.

## Data Availability

All data are available in the supplementary Tables (Tables S1–S3). This study includes no data deposited in external repositories.

## Supplementary Information

## Acknowledgements

We acknowledge Witold Filipowicz for different plasmid constructs. We also acknowledge the support of Malini Sen, Shilpak Chatterjee, and Syamal Roy for different reagents used in ELISA and ATP measurement assays. SN Bhattacharyya and K Mukherjee acknowledge Council of Scientific and Industrial Research (CSIR)-Indian Institute of Chemical Biology (IICB) for infrastructural support. The work is supported by The Swarnajayanti Fellowship (DST/SJF/LSA-03/2014-15) and CSIR funded Niche Creating Project RNAMOD (MLP-139), Leishmaniasis (MLP-136), and the Indo-Swiss Bilateral Project Grant from Department of Biotechnology, Govt of India (BT/IN/Swiss/53/SNB/2018-19). K Mukherjee and SN Bhattacharyya both received support from CSIR. SN Bhattacharyya also acknowledges support of National Bioscience Award Fund from Department of Biotechnology, Government of India. S Ganguly received support from University Grant Commission, India. B Ghoshal, I Banerji, S Chakraborty, S Bhattacharjee, and A Goswami received their support from CSIR, India. The funders had no role in study design, data collection and analysis, decision to publish, or preparation of the manuscript.

## Author Contributions

S Ganguly: data curation, formal analysis, validation, investigation, and methodology.
B Ghoshal: formal analysis, validation, investigation, and methodology.
I Banerji: data curation, formal analysis, validation, investigation, and methodology.
S Bhattacharjee: validation and investigation.
S Chakraborty: validation and investigation.
A Goswami: validation and investigation.
K Mukherjee: conceptualization, formal analysis, supervision, visualization, methodology, and writing—original draft, review, and editing.
SN Bhattacharyya: conceptualization, data curation, supervision, visualization, methodology, project administration, and writing—original draft, review, and editing.

### Conflict of Interest Statement

The authors declare that they have no conflict of interest.

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
