## [Reviewer comments · Life Science Alliance]

Life Science Alliance

Leishmania Survives by Exporting miR-146a from Infected to Resident Cells to Subjugate Inflammation

Satarupa Ganguly, Bartika Ghoshal, Ishani Banerji, Shreya Bhattacharjee, Sreemoyee Chakraborty, Avijit Goswami, Kamalika Mukherjee, and Suvendra Bhattacharyya

DOI: <https://doi.org/10.26508/lsa.202101229>

Corresponding author(s): Suvendra Bhattacharyya, Indian Institute of Chemical Biology and Kamalika Mukherjee, Indian Institute of Chemical Biology

Review Timeline:

Submission Date:	2021-09-08
Editorial Decision:	2021-10-19
Revision Received:	2022-01-20
Editorial Decision:	2022-02-03
Revision Received:	2022-02-07
Accepted:	2022-02-08

Scientific Editor: Novella Guidi

Transaction Report:

October 19, 2021

Re: Life Science Alliance manuscript #LSA-2021-01229-T

Dr. Suvendra N Bhattacharyya
Department of Science and technology, Govt. of India Molecular Genetics
Molecular Genetics Division
RNA Biology Research Laboratory, 4 Raja S C Mullick Road
Kolkata, West Bengal 700032
India

Dear Dr. Bhattacharyya,

Thank you for submitting your manuscript entitled "Leishmania Survives by Cross-Communicating Host miR-146a to Subjugate HuR and miR-122 in Neighbours" to Life Science Alliance. The manuscript was assessed by expert reviewers, whose comments are appended to this letter. As you will note from the reviewers' comments below, both reviewers are quite positive about the study and they both raise few minor issues regarding explanations and few text and figures modifications to address in the revised version. Rev1, besides text and figure modifications, ask to add a positive control to study the mitochondrial dynamics by Ld and to evaluate protein expression levels in addition to mRNA levels of TNF α and IL1 β . Rev 2 requests few text explanations and ask to restructure the results section which is very lengthy and confusing. We, thus, encourage you to submit a revised version of the manuscript back to LSA that responds to all of the reviewers' points.

Thank you for this interesting contribution to Life Science Alliance. We are looking forward to receiving your revised manuscript.

Sincerely,

B. MANUSCRIPT ORGANIZATION AND FORMATTING:

Reviewer #1 (Comments to the Authors (Required)):

Title: Leishmania Hijacks microRNA Import-Export Machinery of Infected Macrophage and Survives by Cross-Communicating Host miR-146a to Subjugate HuR and miR-122 in Neighbouring cells.

In this manuscript, Ganguly et al. reported a novel mechanism by which Leishmania hijacks microRNA import-export machinery of infected macrophages. Also, the authors elegantly demonstrated how Leishmania survives by cross-communicating host miR-146a to subjugate HuR and miR-122 in neighboring cells. Overall, this study is well-designed and written, and the authors used appropriate positive and negative control to validate their experiments. The authors need to address the following comments while revising the manuscript.

Comments:

1. The authors need to discuss how miR-122 targets mitochondrial function.
2. To study the mitochondrial dynamics by Ld, the authors used overexpression of FA-Ucp2 and Mfn2 KO, both reducing ATP levels. The depletion of ATP will affect the endocytosis. In addition to the above controls, I suggest the author use some other positive control.
3. This manuscript has several typo and alignment errors; for example, Fig.7 E has an alignment error in beta-actin, and Fig.3A has a hidden letter. I also suggest authors use the standard abbreviations for reagents.
4. Provide a decent quality western blot image for Ucp2 in Figure 1B.
5. The rigor of the study will be strengthened if the authors provide all the bar graphs showing individual points.
6. In addition to determining the mRNA expression levels of TNF α and IL1 β , the protein expression levels also need to be determined either by western blotting or ELISA.
7. Something is hidden in the labeling of Figure 3A.
8. Some of the Western blot pictures are cropped very closely. The authors need to trim the western blot images with appropriate spacing.
9. Proper labeling is needed for proteins - for example, actin, b-actin, or β -actin? Also, some western blot images lack relative molecular weight labeling.
10. There was an inconsistency in labeling in western blotting results; for example, molecular weight is missing for figures 4, 5, and 6.
11. The abstract needs to be re-written.

Reviewer #2 (Comments to the Authors (Required)):

The manuscript entitled "Leishmania Hijacks microRNA Import-Export Machinery of Infected Macrophage and Survives by Cross-Communicating Host miR-146a to Subjugate HuR and miR-122 in Neighboring cells" aims to identify miRNAs linked survival strategies of Leishmania parasites inside hostile environment of host macrophages. The investigators have identified that Leishmania upregulates miR146a export from infected macrophages to the neighboring naïve macrophages and polarizes them to M2 state that largely produces IL-10, and also inhibit the production of proinflammatory cytokines. In addition, they also reveal that parasite restricts inflammatory mir122 production by liver cells and by altering infected cells mitochondrial functions (depolarization) prevents its entry to infected cells to curtail inflammatory response by infected cells. Although, investigators have done substantial experiments to establish the roles of both miRNAs however, clarifications/explanations/modifications are required on the following points.

1. How did investigators ensure that the polarization of naïve macrophages to M2 state is only because of miR146a, which is coming out from the infected cells. The Leishmania excretory-secretory antigens (LESA), secreted from the leishmania infected macrophages or other antigen/s, may also alter macrophage's function and their polarizations. The LESA is already known to activate naïve macrophages to produce anti-inflammatory cytokines and prime naïve cells for their smooth entry, which may be a reason for the increased IL-10 production and their polarization into M2 state. This requires a proper explanation.

2. Did investigators measure miR122 levels in Leishmania antigens (SLA/LESA) activated macrophages prior to mir122 EV treatment in order to verify if it is produced by macrophages as well? Similarly, the generation of miR146a by hepatocytes in response to parasitic antigens should have also been analyzed by the investigators to ensure that it is the trafficking of EVs that regulates effector functions of naïve and infected macrophages not the Leishmania proteins/antigens.
3. The authors should also provide a proper justification that why they quantified TNF-a (most of the times) levels as a proinflammatory marker in infected macrophages. It is well known that NOx is primary anti-Leishmania molecule of macrophages, which production is more linked with IL-12 levels in the infected cells.
4. Although, the investigators have provided enough experimental data on mitochondrial depolarization by Leishmania but it is not clear that how does it stop miR122 containing EVs entry to macrophages? There are other micro-RNAs which provide stability to the mitochondria and thus balance the detrimental effect of infections. Thus, a brief explanation is needed in support that only Ucp2 expression can inhibit the entry of miR-122.
5. The results are not well presented. The whole section is very lengthy and confusing. There are plenty of statements, which are irrelevantly placed through the text. For e.g. "The infected cells prevent the miR-122 containing EVs entry and restricts the activation of the Ld-infected cells by miR-122 in order to prevent the death of the internalized parasites due to enhanced pro-inflammatory response observed in miR-122 recipient macrophage (Chen et al, 2011)". I could not understand the purpose of this statement and the citation as well. There are plenty of such statements. Thus, I will recommend authors to fix such errors to make statements clear and focused. Also, authors are advised to send a few main figures (Like Figure 1, which is not clearly presented as well, Figure 5 as TRAF6, MyD88, p38, TNF-a, IL-10 linked is already well known in Leishmania infection) to the supplementary info.
6. The role of HuR in induction of IL-10 needs to be further discussed as it can be seen in Fig. 7D and 7F that HuR silencing does not significantly decreases IL-10.
7. The manuscript have several syntax errors.

Answers to Reviewers' comments

Reviewer #1 (Comments to the Authors (Required)):

Title: Leishmania Hijacks microRNA Import-Export Machinery of Infected Macrophage and Survives by Cross-Communicating Host miR-146a to Subjugate HuR and miR-122 in Neighbouring cells.

In this manuscript, Ganguly et al. reported a novel mechanism by which Leishmania hijacks microRNA import-export machinery of infected macrophages. Also, the authors elegantly demonstrated how Leishmania survives by cross-communicating host miR-146a to subjugate HuR and miR-122 in neighboring cells. Overall, this study is well-designed and written, and the authors used appropriate positive and negative control to validate their experiments. The authors need to address the following comments while revising the manuscript.

Comments:

1. The authors need to discuss how miR-122 targets mitochondrial function.

This is an interesting question to address. miR-122 is known to regulate mitochondrial metabolic function. Cationic amino acid transporter gene CAT-1 level decreases while PPARGC1A (PGC-1 α) and Succinate dehydrogenase subunit A & B levels increase in HCC cells expressing miR-122. PGC-1 α is the regulator of mitochondrial biogenesis and it is also involved in energy metabolism. Succinate dehydrogenase (SDH), the enzyme complex located in inner mitochondrial membrane, is involved in both TCA cycle & electron transport chain. PGC-1 α & Succinate dehydrogenase both are found to be the putative secondary targets of miR-122 (Burchard et al 2010). To find out the effect of miR-122 on mitochondrial function we expressed miR-122 exogenously in RAW264.7 cells by transfecting precursor miR-122 expression plasmid pmiR-122. Mitochondrial depolarization as well as ATP content of pmiR-122 transfected cells were measured. Interestingly we found that there has been an increase in ATP content in miR-122 expressing RAW264.7 cells (New Figure S3E) however

change observed in mitochondrial polarization due to miR-122 expression has not been significant (New Figure S3F). We have mentioned these findings in the discussion section of the manuscript.

2. To study the mitochondrial dynamics by Ld, the authors used overexpression of FA-Ucp2 and Mfn2 KO, both reducing ATP levels. The depletion of ATP will affect the endocytosis. In addition to the above controls, I suggest the author use some other positive control.

It has been reported previously that mitochondrial tethering with ER in mammalian cells is important for miRNA compartmentalization and a failed ER-mitochondria tethering resulted in defective miRNA transport and recycling that what exactly happens in *Ld*-infected cells where the mitochondrial tethering defects causes the retarded miRNA recycling (Bose et al 2020, Chakrabarty & Bhattacharyya 2017, Chatterjee et al 2020). In our previous work we have found how the Mitochondria-ER targeting and endosome dynamics play a role in miRNA compartmentalization process and thus must be affected by any factors that could affect organellar dynamics in mammalian cells. Increased expression of Ucp2 due to *Ld* infection or exogenous expression of FH-Ucp2 is associated with mitochondrial depolarization and changed mitochondrial dynamics that resulted in miRNA compartmentalization defect. Importantly Mfn2 loss in *Ld*-infected cells or in Mfn2 KO cells are also associated with miRNA compartmentalization defect (Chakrabarty & Bhattacharyya 2017). Therefore, microRNA compartmentalization is possibly got affected by changes in mitochondrial dynamics and ER-tethering rather than changed membrane potential or ATP level. We have designed a separate set of experiments where we have treated recipient RAW264.7 cells with *p*-Trifluoromethoxyphenylhydrazine (FCCP) (0.5 μ M) or Oligomycin (2.5 μ M) for 3hours followed by miR-122 containing EV treatment for 16 hours. DMSO or Ethanol treatment was used as controls for FCCP or Oligomycin respectively. It is known already that FCCP is an uncoupler of mitochondrial membrane potential but does not affect

cellular ATP content whereas oligomycin inhibits F₀F₁-ATPase and prevents ATP production without affecting mitochondrial membrane potential (Chakrabarty & Bhattacharyya 2017). In our experiment, we found that oligomycin treatment could not inhibit EV-mediated internalization of miR-122 and there was also no change in miR-122 uptake in 3h FCCP treated cells compared to the DMSO control (New Figure S3C). Thus, rather than mitochondrial depolarization or ATP depletion it is the change in mitochondrial dynamics and interaction with endosomes and ER in Mfn2 KO or FH-Ucp2 expressing or *Ld*-infected cells(Chakrabarty & Bhattacharyya 2017) that could probably hampers endocytosis of miR-122 containing EVs in macrophage cells. We have included these results and address the issue in the discussion part of the revised manuscript.

3. This manuscript has several typo and alignment errors; for example, Fig.7 E has an alignment error in beta-actin, and Fig.3A has a hidden letter. I also suggest authors use the standard abbreviations for reagents.

We apologise for such mistakes, and we have tried to correct all possible errors.

4. Provide a decent quality western blot image for Ucp2 in Figure 1B.

We have included new western blot image for Ucp2 for Figure 1B as required. Also, we have given a graphical representation of densitometric analysis of Ucp2 western blots.

5. The rigor of the study will be strengthened if the authors provide all the bar graphs showing individual points.

Following the suggestions, we have changed the graphical representation of data now showing all data points.

6. In addition to determining the mRNA expression levels of TNF α and IL1 β , the protein expression levels also need to be determined either by western blotting or ELISA.

Throughout the manuscript we have noted induction of TNF- α and IL-1 β mRNAs in macrophage or RAW264.7 cells expressing or receiving miR-122 while the expression or uptake of miR-146 is associated with a reduction of the same in RAW264.7 or macrophage cells. To test the expression and detection of these cytokines at protein level, we have done ELISA assays with the culture supernatants in RAW264.7 cells expressing miR-122 or miR-146a. LPS stimulated macrophage culture supernatant were used as positive controls. We observe an increase in TNF- α protein level in LPS treated as well as miR-122 overexpressed condition whereas in miR-146a overexpressed condition a sharp decrease in TNF- α level has been observed (New Figure 1C).

In miR-146a overexpressing condition, as expected, we detect decrease in IL-1 β protein level in cell lysates compared to control cells (Figure S6D). However, we could not detect an increase in mature IL-1 β by ELISA in miR-122 expressing cells. Regarding the detection of IL-1 β , it has been reported earlier that IL-1 β produced in an inactive precursor pro- IL-1 β form and this induction by PAMPs is an initial priming step which cannot trigger its secretion. After the priming stimulus, cell required other stimuli which induce the processing & secretion of precursor of IL-1 β . Usually, inflammasome mediated activation of caspase-1 is responsible for the processing of Pro- IL-1 β to mature form followed by secretion (Lopez-Castejon & Brough 2011, Martinon et al 2002). Thus, the transcriptional surge of IL-1 β mRNA due to miR-122 expression may not be associated with increase in the mature IL-1 β protein detectable by ELISA. It is also possible that the excess IL-1 β mRNA produced in presence of miR-122 may get transferred to other immune cells where they may elicit a proinflammatory response rather than in the originating macrophage cells. Therefore,

inflammatory response observed in miR-122 expressing cells is possibly manifested primarily by TNF- α rather than IL-1 β . We have discussed this aspect in the discussion section of the manuscript.

7. Something is hidden in the labeling of Figure 3A.

We are sorry for the technical problem where the annotation being partly masked by the image itself and now has been taken care of.

8. Some of the Western blot pictures are cropped very closely. The authors need to trim the western blot images with appropriate spacing.

We have paid particular attention to this issue and have presented some of the blot images with appropriate image length and size. Original blots data are also provided with the revised version.

9. Proper labeling is needed for proteins - for example, actin, b-actin, or β -actin? Also, some western blot images lack relative molecular weight labelling.

These errors have been rectified and specific attention has been given to the incorporate the molecular weight labelling for each blot.

10. There was an inconsistency in labeling in western blotting results; for example, molecular weight is missing for figures 4, 5, and 6.

We have rectified the errors.

11. The abstract needs to be re-written.

We have rewritten the abstract to certain extent to make it more communicative.

Reviewer #2 (Comments to the Authors (Required)):

The manuscript entitled "Leishmania Hijacks microRNA Import-Export Machinery of Infected Macrophage and Survives by Cross-Communicating Host miR-146a to Subjugate HuR and miR-122 in Neighboring cells" aims to identify miRNAs linked survival strategies of Leishmania parasites inside hostile environment of host macrophages. The investigators have identified that Leishmania upregulates miR146a export from infected macrophages to the neighboring naïve macrophages and polarizes them to M2 state that largely produces IL-10, and also inhibit the production of proinflammatory cytokines. In addition, they also reveal that parasite restricts inflammatory mir122 production by liver cells and by altering infected cells mitochondrial functions (depolarization) prevents its entry to infected cells to curtail inflammatory response by infected cells. Although, investigators have done substantial experiments to establish the roles of both miRNAs however, clarifications/explanations/modifications are required on the following points.

1. How did investigators ensure that the polarization of naïve macrophages to M2 state is only because of miR146a, which is coming out from the infected cells. The Leishmania excretory-secretory antigens (LESA), secreted from the leishmania infected macrophages or other antigen/s, may also alter macrophage's function and their polarizations. The LESA is already known to activate naïve macrophages to produce anti-inflammatory cytokines and prime naïve cells for their smooth entry, which may be a reason for the increased IL-10 production and their polarization into M2 state. This requires a proper explanation.

It has been known that miR-146a favours the anti-inflammation and is associated with M2 polarization of macrophages (Huang et al 2016). Also, the role of LESA in IL-10 production is also known as mentioned by the reviewer. To understand the role of EV mediated transfer of

miR-146a on M2 state of macrophages, we overexpress the precursor form of the miR-146a by transfecting naïve RAW264.7 cells with Pre-miR-146a expression plasmid and isolated EVs were used to treat naïve macrophages. Figure 6H showed the increased mRNA level of the anti-inflammatory cytokine IL-10 & decreased pro-inflammatory cytokine IL-1 β mRNA level in miR-146a containing-EV treated macrophages. Thus miR-146a containing EV treated macrophages showed the similar cytokine pattern like the infected cell derived EV treated macrophages polarized to M2 as described in Figure 6F. In our experimental system, the effect of Leishmanial secretory proteins on the M2 state of macrophages is seems to be marginal as the cytokine pattern of recipient macrophages treated with promastigote culture derived EVs (as shown in Figure S7C) showed decreased level of IL-10 & IL-1 β as well. Thus, these results clearly explains that the observed M2 state of the naïve macrophages treated with infected cell derived EVs is due to EV mediated transfer of miR-146a and possibly not by LESA. This received further support from the data we obtained with Leishmania derived SLA or LPG discussed below. We have also included a discussion on these results in the relevant part of the manuscript.

2. Did investigators measure miR122 levels in Leishmania antigens (SLA/LESA) activated macrophages prior to mir122 EV treatment in order to verify if it is produced by macrophages as well? Similarly, the generation of miR146a by hepatocytes in response to parasitic antigens should have also been analyzed by the investigators to ensure that it is the trafficking of EVs that regulates effector functions of naïve and infected macrophages not the Leishmania proteins/antigens.

This is an important concern, and we appreciate the reviewers view on that. To determine whether Leishmania derived factors can induce miR-122 production in macrophages prior EV treatment, we prepared Soluble Leishmanial Antigens (SLA) from stationary phase

promastigote culture and treated macrophages for 24 hours with 10µg of SLA to determine the level of miR-122 there. Although, it is known that macrophages expressed negligible level of miR-122 compared to hepatocytes (Aucher et al 2013). We found no increase in miR-122 level in treated macrophages (New Figure S7E). When macrophages were treated with parasite derived EVs or LPG, there have been no increase in miR-122 level also (New Figure S7D and G).

To ensure that the increased level of miR-146a in recipient hepatocytes is mediated primarily by infected macrophage derived EV and not by Leishmanial proteins or antigens, we treated hepatocytes for 24h with 10µg of SLA to determine the changes in the level of miR-146a. Treated hepatocytes showed reduced level of miR-146a (New Figure S7E). Parasite derived EV or LPG treated hepatocytes also showed reduced level of miR-146a (New Figure S7D and G). Thus, results obtained from the SLA or LPG or parasite derived EV treated macrophages & hepatocytes suggest Leishmania derived factors are not primarily responsible for miR-122 & miR-146a upregulation in respective cell types.

3. The authors should also provide a proper justification that why they quantified TNF- α (most of the times) levels as a proinflammatory marker in infected macrophages. It is well known that NOx is primary anti-Leishmania molecule of macrophages, which production is more linked with IL-12 levels in the infected cells.

We are totally agreed with the reviewer that NOx is the primary anti-Leishmanial molecule released by macrophages and linked to IL-12. It is also known that leishmania infection reduces IL-12. However, there are reports which showed that TNF- α mediates Nitric oxide generation in macrophages preventing *Leishmania major* infection (Fonseca et al 2003, Silva et al 1995). Also, anti- TNF- α antibody treated mice showed decrease NO production and higher parasite load in *Trypanosoma cruzi* infection (Silva et al 1995). In fact, in our previous study, we did observed a downregulation of TNF- α mRNA in macrophage after 24

hours of *Leishmania donovani* infection (Goswami et al 2020). Thus, we considered TNF- α cytokine as one of the pro-inflammatory markers that get downregulated in infected macrophages.

Following the notion of involvement of NO in inflammatory pathway in macrophage cells, we also checked the cellular level of iNOS mRNA as well as Nitric oxide level in culture supernatant in miR-122 & miR-146a expressing macrophages. As noted earlier, expression of miR-122 induces TNF- α mRNA in macrophage (Figure 1C) while expression of miR-146a reduces TNF- α mRNA (Figure 6I). Here, we also checked iNOS mRNA level as well as Nitric oxide level in miR-122 & miR-146a expressing cells. Our result revealed increased level of iNOS mRNA in miR-122 expressing macrophage as it happened in LPS treated macrophages (100ng/ml) (Figure 1C) whereas decreased iNOS level were observed in miR-146a expressing cells (Figure 1C). Nitric oxide level showed similar increase in culture supernatant of LPS treated or miR-122 expressing macrophages. The decrease in NO was noted in miR-146a expressing cell supernatant (Figure 1C). This data suggests possible synergy in TNF- α & NO generation to induce anti-inflammatory responses against *Ld* infection.

4. Although, the investigators have provided enough experimental data on mitochondrial depolarization by Leishmania but it is not clear that how does it stop miR122 containing EVs entry to macrophages? There are other micro-RNAs which provide stability to the mitochondria and thus balance the detrimental effect of infections. Thus, a brief explanation is needed in support that only Ucp2 expression can inhibit the entry of miR-122.

In Figure 3B, expression of FH-Ucp2 in recipient HeLa cells reduces internalization of miR-122 containing EVs. As a result of which, we have found reduced level of miR-122 in the recipient HeLa cells expressing FH-Ucp2 compared to the control cells treated with miR-122

containing EVs as shown in Figure 3C. Along with the miR-122 level, we checked the activity of miR-122 in recipient HeLa cells by following the repression of miR-122 site containing luciferase reporter mRNA and noted a reduced activity of miR-122 in FH-Ucp2 expressing recipient HeLa cells (Figure 3D). While HeLa cells does not express miR-122, naïve RAW264.7 cells express only a basal level of miR-122. We observed expression of FH-Ucp2 reduces endogenous miR-122 level (New Figure S3D) in RAW264.7 cells. This result suggest that FH-Ucp2 expression reduces the entry of miR-122 via EV and also the endogenous miR-122 production in RAW264.7 cells. This has been discussed in the relevant section of the manuscript.

We have also mentioned in our reply to Reviewer1, Point 2 that loss of mitochondrial dynamics noted both in *Ld*-infected and FH-Ucp2 expressing cells (Chakrabarty & Bhattacharyya 2017), is the cause an altered endoplasmic reticulum and endosome interaction. This interaction loss is supposed to cause a defect in miRNA recycling (Bose et al 2020) and defective miRNA entry via EVs and is independent of mitochondrial polarization status and cellular ATP concentration. This has also been discussed in the revised manuscript.

5. The results are not well presented. The whole section is very lengthy and confusing. There are plenty of statements, which are irrelevantly placed through the text. For e.g. "The infected cells prevent the miR-122 containing EVs entry and restricts the activation of the Ld-infected cells by miR-122 in order to prevent the death of the internalized parasites due to enhanced pro-inflammatory response observed in miR-122 recipient macrophage (Chen et al, 2011)". I could not understand the purpose of this statement and the citation as well. There are plenty of such statements. Thus, I will recommend authors to fix such errors to make statements clear and focused. Also, authors are advised to send a few main figures (Like Figure

1, which is not clearly presented as well, Figure 5 as TRAF6, MyD88, p38, TNF- α , IL-10 linked is already well known in Leishmania infection) to the supplementary info.

Following the suggestion and well taken advise of the reviewer we have reshuffled some statements and sentences that were otherwise confusing. As it carries important information related to *Ld* infection and M2 polarization by miR-146a in macrophage cells, we have moved the Supplementary Figure 8 as a main Figure (New Figure 8) now.

6. The role of HuR in induction of IL-10 needs to be further discussed as it can be seen in Fig. 7D and 7F that HuR silencing does not significantly decreases IL-10.

IL-10 is regulated by HuR. By exporting miR-21, HuR ensures derepression of IL-10 in cells also expressing the IL-10 repressor miR-21. Therefore, as expected the HA-HuR expression should increase the IL-10 production as noted in Figure 7D. However, the siHuR treatment has a minimal effect on IL-10 level (Figure 7F). This may be contributed by excess miR-146 present in the cells treated with miR-146a containing EVs. HuR undergo two separate levels of regulation in presence of miR-146a in macrophage. In initial phase the high HuR facilitate the IL-10 expression but on contrary it also stabilizes the IL-1 β and other proinflammatory cytokines having AU-rich elements in their 3' UTR. This in turn affects the inflammatory responses. It has been noted that the excess miR-146a also reduces HuR while upregulating the IL-10 (Figure 7B, H). It is possible that the excess IL-10 has a negative feedback effect on HuR itself to lower HuR for IL-10 upregulation in presence of miR-146a. Therefore, a biphasic regulation of inflammatory response by HuR in presence of miR-146a has been envisioned in this manuscript where in initial phase of M2 polarization the macrophage must have HuR in place to get the IL-10 expressed in excess. IL-10 in turn lowers HuR to restrict the HuR mediated stabilization and expression of IL-1 β and TNF- α mRNAs with AU-rich elements.

7. The manuscript have several syntax errors.

We have tried to fix all such errors to improve the quality of the manuscript.

References:

Aucher A, Rudnicka D, Davis DM. 2013. MicroRNAs transfer from human macrophages to hepatocarcinoma cells and inhibit proliferation. *J Immunol.* 191(12):6250-6260. doi:10.4049/jimmunol.1301728

Bose M, Chatterjee S, Chakrabarty Y, Barman B, Bhattacharyya SN. 2020. Retrograde trafficking of argonaute 2 acts as a rate-limiting step for de novo mirnp formation on endoplasmic reticulum-attached polysomes in mammalian cells. *Life science alliance.* 3(2) doi:10.26508/lsa.201800161

Burchard J, Zhang C, Liu AM, Poon RT, Lee NP, Wong KF, Sham PC, Lam BY, Ferguson MD, Tokiwa G, et al. 2010. MicroRNA-122 as a regulator of mitochondrial metabolic gene network in hepatocellular carcinoma. *Molecular systems biology.* 6:402. doi:10.1038/msb.2010.58

Chakrabarty Y, Bhattacharyya SN. 2017. *Leishmania donovani* restricts mitochondrial dynamics to enhance mirnp stability and target rna repression in host macrophages. *Molecular biology of the cell.* doi:10.1091/mbc.E16-06-0388

Chatterjee S, Chakrabarty Y, Banerjee S, Ghosh S, Bhattacharyya SN. 2020. Mitochondria control mtorc1 activity-linked compartmentalization of eif4e to regulate extracellular export of microRNAs. *Journal of cell science.* 133(24) doi:10.1242/jcs.250241

Fonseca SG, Romao PR, Figueiredo F, Morais RH, Lima HC, Ferreira SH, Cunha FQ. 2003. Tnf-alpha mediates the induction of nitric oxide synthase in macrophages but not in neutrophils in experimental cutaneous leishmaniasis. *European journal of immunology.* 33(8):2297-2306. doi:10.1002/eji.200320335

Goswami A, Mukherjee K, Mazumder A, Ganguly S, Mukherjee I, Chakrabarti S, Roy S, Sundar S, Chattopadhyay K, Bhattacharyya SN. 2020. MicroRNA exporter hur clears the internalized pathogens

by promoting pro-inflammatory response in infected macrophages. *EMBO molecular medicine*. 12(3):e11011. doi:10.15252/emmm.201911011

Huang C, Liu XJ, QunZhou, Xie J, Ma TT, Meng XM, Li J. 2016. Mir-146a modulates macrophage polarization by inhibiting notch1 pathway in raw264.7 macrophages. *International immunopharmacology*. 32:46-54. doi:10.1016/j.intimp.2016.01.009

Lopez-Castejon G, Brough D. 2011. Understanding the mechanism of il-1beta secretion. *Cytokine & growth factor reviews*. 22(4):189-195. doi:10.1016/j.cytogfr.2011.10.001

Martinon F, Burns K, Tschopp J. 2002. The inflammasome: A molecular platform triggering activation of inflammatory caspases and processing of proil-beta. *Molecular cell*. 10(2):417-426. doi:10.1016/s1097-2765(02)00599-3

Silva JS, Vespa GN, Cardoso MA, Aliberti JC, Cunha FQ. 1995. Tumor necrosis factor alpha mediates resistance to trypanosoma cruzi infection in mice by inducing nitric oxide production in infected gamma interferon-activated macrophages. *Infection and immunity*. 63(12):4862-4867. doi:10.1128/iai.63.12.4862-4867.1995

February 3, 2022

RE: Life Science Alliance Manuscript #LSA-2021-01229-TR

Dr. Suvendra N Bhattacharyya
Indian Institute of Chemical Biology
Molecular Genetics Division
RNA Biology Research Laboratory, 4 Raja S C Mullick Road
Kolkata, West Bengal 700032
India

Dear Dr. Bhattacharyya,

Thank you for submitting your revised manuscript entitled "Leishmania Survives by Exporting miR-146a from Infected to Resident Cells to Subjugate Inflammation". We would be happy to publish your paper in Life Science Alliance pending final revisions necessary to meet our formatting guidelines.

- please add ORCID ID for secondary corresponding author-they should have received instructions on how to do so
- please add the Twitter handle of your host institute/organization as well as your own or/and one of the authors in our system
- please consult our manuscript preparation guidelines <https://www.life-science-alliance.org/manuscript-prep> and make sure your manuscript sections are in the correct order
- please add a conflict of interest statement to your main manuscript text
- please correct the label for figure 8
- please add a callout for Figure S4 to your main manuscript text
- please add a separate Data Availability section including the mass spectrometry data

FIGURE CHECKS:

- there is a strange line in figure S2C RILP: please provide source data for this figure
- 3rd blot from the left in S5H beta-actin looks like it doesn't belong to the continuous blots: please provide source data for this figure as well

A. FINAL FILES:

B. MANUSCRIPT ORGANIZATION AND FORMATTING:

Sincerely,

Reviewer #1 (Comments to the Authors (Required)):

the revision has strengthened the manuscript significantly and is acceptable for publication!

Reviewer #2 (Comments to the Authors (Required)):

The authors have made all the necessary changes as per comments. I am satisfied with the quality of the manuscript in its current form, and thus recommend its publication in LSA.

February 8, 2022

RE: Life Science Alliance Manuscript #LSA-2021-01229-TRR

Dr. Suvendra N Bhattacharyya
Indian Institute of Chemical Biology
Molecular Genetics Division
RNA Biology Research Laboratory, 4 Raja S C Mullick Road
Kolkata, West Bengal 700032
India

Dear Dr. Bhattacharyya,

Thank you for submitting your Research Article entitled "Leishmania Survives by Exporting miR-146a from Infected to Resident Cells to Subjugate Inflammation". It is a pleasure to let you know that your manuscript is now accepted for publication in Life Science Alliance. Congratulations on this interesting work.

DISTRIBUTION OF MATERIALS:

Again, congratulations on a very nice paper. I hope you found the review process to be constructive and are pleased with how the manuscript was handled editorially. We look forward to future exciting submissions from your lab.

Sincerely,
